# Segment-specific axon guidance by Wnt/Fz signaling diversifies motor commands in *Drosophila* larvae

**Suguru Takagi[1†], Shiina Takano[1], Tomohiro Kubo[2], Yusaku Hashimoto[1], Shu Morise[1], Xiangsunze Zeng[1], Akinao Nose[1,2]***

[1]Department of Complexity Science and Engineering, Graduate School of Frontier Sciences, The University of Tokyo, Tokyo, Japan; [2]Department of Physics, Graduate School of Science, The University of Tokyo, Tokyo, Japan

**\*For correspondence:**
nose@k.u-tokyo.ac.jp

**Present address:** [†]Center for Integrative Genomics, Faculty of Biology and Medicine, University of Lausanne, Lausanne, Switzerland

**Competing interest:** The authors declare that no competing interests exist.

## eLife Assessment

The study by Takagi and colleagues is an **important** contribution to the question of how homologous neuronal circuits might be wired differently to elicit specific behaviours. The authors combine genetic, neuroanatomical, and behavioral data to provide **convincing** evidence that Dfz2/DWnt4 signaling controls the innervation pattern of wave command neurons in the fly larva, and thereby behavioral locomotion program selection.

**Abstract** Functional diversification of homologous neuronal microcircuits is a widespread feature observed across brain regions, as well as across species, while its molecular and developmental mechanisms remain largely unknown. We address this question in *Drosophila* larvae by focusing on segmentally homologous Wave command-like neurons, which diversify their wiring and function in a segment-specific manner. Anterior Wave (a-Wave) neurons extend axons anteriorly and connect to circuits inducing backward locomotion, whereas posterior Wave (p-Wave) neurons extend axons posteriorly and trigger forward locomotion. Here, we show that Frizzled receptors DFz2 and DFz4, together with the DWnt4 ligand, regulate the segment-specific Wave axon projection. *DFz2* knockdown (KD) not only reroutes Wave axons to posterior neuromeres but also biases its motor command to induce forward instead of backward locomotion as tactile response. Thus, segment-specific axon guidance diversifies the function of homologous command neurons in behavioral regulation. Since control of anterior-posterior (A-P) axon guidance by Wnt/Fz signaling is evolutionarily conserved, our results reveal a potentially universal molecular principle for formation and diversification of the command system in the nerve cord. Furthermore, this work indicates that sensorimotor transduction can be rerouted by manipulating a single gene in a single class of neurons, potentially facilitating the evolutionary flexibility in action selection.

## Introduction

The adaptive behavior of an animal is underpinned by the complex, yet precise, connectivity of its nervous system. For ecologically fit behaviors to occur, information regarding various sensory inputs must be routed to appropriate motor outputs by the neuronal circuits, which are established during development. Previous studies on axon guidance and selective synapse formation revealed the mechanisms of how connectivity between a pair of neurons is formed during development (*Kolodkin and Tessier-Lavigne, 2011*; *Tessier-Lavigne and Goodman, 1996*; *Tosches, 2017*). However, how a

population of neurons as a whole is wired to form functional circuits and generate adaptive behaviors remains unknown.

Functional diversification of homologous brain regions is a widespread phenomenon observed across brain regions and across species (*Chakraborty and Jarvis, 2015*; *Tosches, 2017*), including spinal segments (*Leung and Shimeld, 2019*; *Levine et al., 2012*), basal ganglia (*Grillner and Robertson, 2016*), cerebral (*Tosches et al., 2018*), and cerebellar (*Kebschull et al., 2020*) cortices. For instance, the spinal cord in vertebrates and ventral nerve cord (VNC) in invertebrates are composed of homologous neuromeres, which link inputs from the corresponding body segment to appropriate motor output(s) (*Barthélemy et al., 2006*; *Levine et al., 2014*; *Saltiel et al., 1998*; *Takagi et al., 2017*; *Tresch and Bizzi, 1999*). Accordingly, adaptive behaviors may be established in part by diversifying the connectivity among homologous circuits, through changes in the circuit's wiring during development. However, whether such a mechanism plays a role in diversifying behaviors remains unclear (*Chakraborty and Jarvis, 2015*; *Katz, 2011*; *Kirschner and Gerhart, 1998*; *Tosches, 2017*). Since the circuit wiring process recruits stepwise interactions with various surrounding cells during development, as well as synaptic matching among a number of pre- and postsynaptic cells, circuit diversification may require orchestrated changes in the expression of a large number of genes across cell types. Alternatively, there may be some degree of flexibility in the system such that modulation of a small number of developmental properties (e.g. changes in the expression of neurite guidance molecules in a small population of cells) is sufficient to alter an animal's behavior. It has been difficult to examine these possibilities due to the lack of tools to visualize and manipulate neural wiring in homologous circuits and our limited knowledge on the relationship between neurite guidance and behavioral regulation.

Previously, we identified a class of segmentally homologous command-like neurons in the VNC of *Drosophila* larvae, named Wave neurons (*Takagi et al., 2017*), which adaptively link tactile inputs from different segments to appropriate motor outputs (*Figure 1A and A'*) by diversifying their connectivity. Wave neurons are present in identical positions in each abdominal neuromere, A1-A7, and extend their proximal axons to the neuropile in similar manners (*Figure 1A–C*). In contrast to such homology, there is a striking difference in their longitudinal axon projection upon entering the neuropile and subsequent formation of functional connectivity. Wave neurons in anterior segments (a-Wave; in neuromeres A1-A3) extend axons anteriorly and connect to circuits inducing backward locomotion, whereas Wave neurons in posterior segments (p-Wave; in neuromeres A4-A7) extend axons posteriorly and elicit forward locomotion. However, which molecular mechanism(s) underlie the diverged Wave A-P axon projections and how they are relevant to behavioral regulations remained unknown.

## Results
### DFz2 and DFz4 regulate segment-specific Wave axon projection

We focused on Wnt receptors (Frizzled/Ryk) as they were known to be involved in bidirectional A-P axon guidance in mammals and nematodes (*Hilliard and Bargmann, 2006*; *Kirszenblat et al., 2011*; *Lyuksyutova et al., 2003*; *Salinas and Zou, 2008*). We performed an RNAi-based KD of *Drosophila* Wnt receptors (*DFz, DFz2, DFz3, DFz4, drl, Drl-2, smo, Corin*) in Wave neurons (see Materials and methods details) and observed their neurite morphology using a GAL4 line that consistently targets Wave neurons from embryonic to larval stages (*Figure 1—figure supplement 1*). We identified two genes, *DFz2* and *DFz4*, as being possibly involved in the axon extension toward the posterior end. *DFz2* KD elongated the axon extension toward the posterior end, whereas *DFz4* KD shortened it (*Figure 1—figure supplement 2A–D*, p=2.38 × 10$^{-6}$, Chi-square test).

To further characterize the role of these receptors, we examined the morphology of single Wave neurons by using MultiColor FlpOut (MCFO, *Nern et al., 2015*) technique while knocking down these receptors in Wave neurons. We found that axon extension toward the posterior end (but not toward the anterior end) was elongated in both a-Wave and p-Wave neurons upon *DFz2* KD by using two independent RNAi lines (A2-Wave to A6-Wave; *Figure 1B–M*; *Figure 1—figure supplement 2F and F'*). Notably, the abnormal extension of Wave axons toward the posterior end was accompanied by presynaptic varicosities en route (*Figure 1D and E*), suggesting that ectopic synapses are formed in this region. The ectopic axon extension from a-Wave neurons intruded the region where p-Wave axons normally project to (*Figure 1F–J*), raising the possibility that a-Wave neurons gain connections

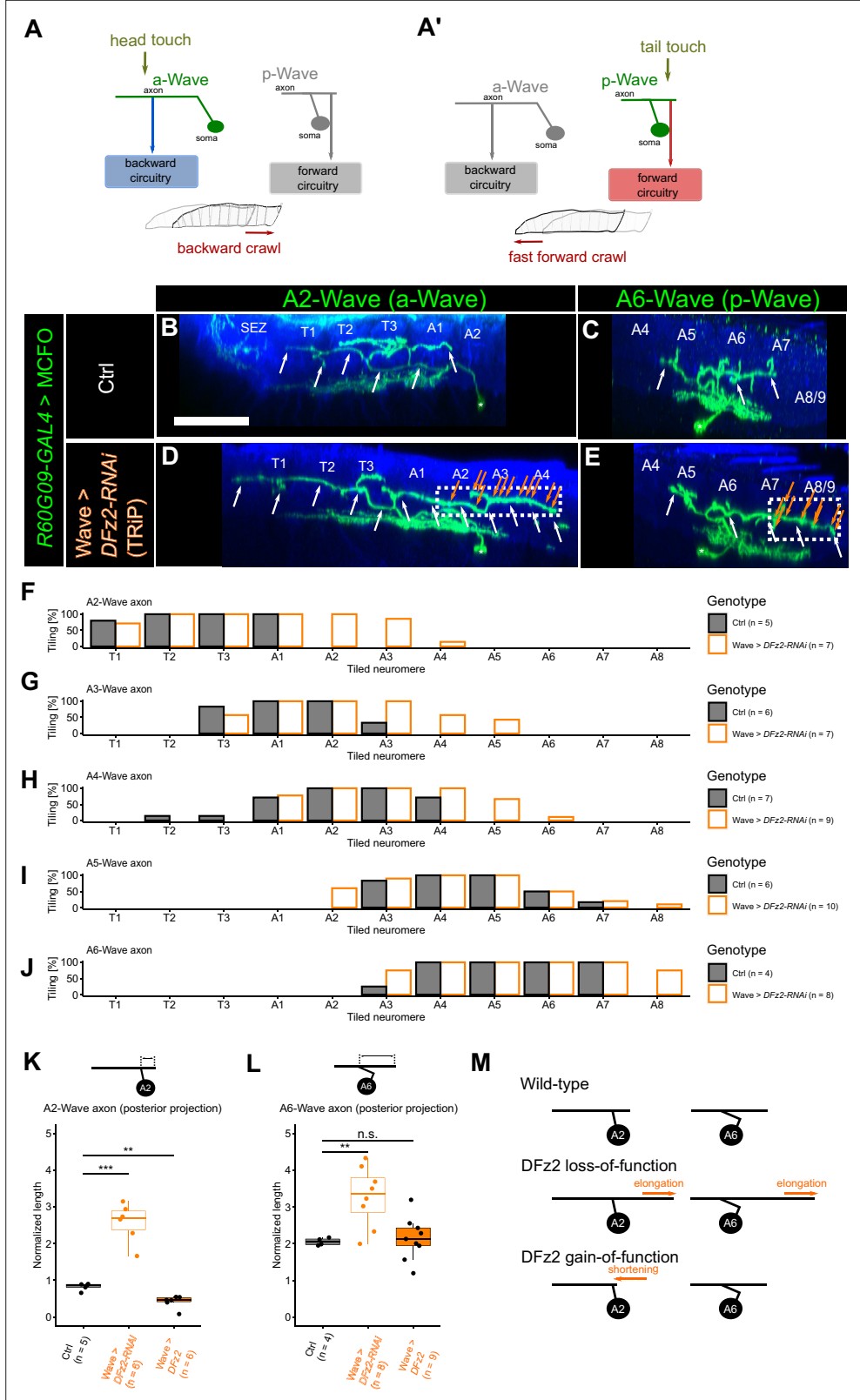

**Figure 1.** DFz2 inhibits Wave axon extension toward the posterior end. (**A, A'**) Schematic diagram of Wave neuron morphology and function. Wave neurons are segmentally repeated in the ventral nerve cord (VNC), receive inputs from tactile sensory neurons, and drive distinct behaviors depending on the segment. The anterior Wave (a-Wave) comprises anteriorly polarized axon/dendrite and drives backward locomotion, whereas posterior Wave

*Figure 1 continued on next page*

*Figure 1 continued*

(p-Wave) has axon/dendrite extension toward the posterior end and drives forward locomotion. (**B–E**) Lateral view of single-cell images of Wave neurons revealed by mosaic analyses in 3rd instar larvae. The green channel shows Wave neurons, and the blue channel shows anti-HRP staining that visualizes the neuropil. SEZ: subesophageal zone, T1-3: thoracic VNC, A1-8/9: abdominal VNC. (**B, C**) Control A2 (**B**) and A6 (**C**) Wave neuron. Scale bar = 50 μm. White arrows indicate axon processes. (**D, E**) A2 (**D**) and A6 (**E**) Wave neuron in which *DFz2* is knocked down using the TRiP RNAi (#67863) line. Note the elongation of the axons toward the posterior end (dotted boxes). Orange arrows indicate the putative presynaptic varicosities in the ectopic axons. (**F–J**) Tiling profile of the axons of A2 to A6 Wave neurons in control and *DFz2* knocked-down animals. *n* indicates the number of neurons. Tiling percentage indicates the fraction of samples in which the Wave axon innervates the corresponding neuromere. (**K, L**) Quantification of axon extension toward the posterior end in A2 (**K**) and A6 (**L**) Wave neuron measured in MultiColor FlpOut (MCFO) images, following knockdown (KD) and overexpression of *DFz2*, respectively. **\*\***: p<0.01, **\*\*\***: p<0.001, Welch's t-test with Bonferroni correction. (**M**) Summary of *DFz2* KD and overexpression phenotypes.

The online version of this article includes the following figure supplement(s) for figure 1:

**Figure supplement 1.** *R60G09-GAL4* consistently targets Wave neurons from embryonic to larval stages.

**Figure supplement 2.** Identification of candidate Wnt receptors and ligands that regulate anterior-posterior (A-P) Wave axon guidance.

**Figure supplement 3.** Effect of *DFz2* knockdown (KD) on Wave dendrite extension.

to the circuits inducing forward locomotion as p-Wave neurons normally do. Conversely, overexpression of *DFz2* in Wave neurons resulted in the shortening of a-Wave (but not p-Wave) axon extension toward the posterior end (*Figure 1K and L*). In summary, DFz2 is necessary for the suppression of Wave neurite outgrowth toward the posterior end (*Figure 1M*), and such suppression might be indispensable to diversify Wave neuron connectivity across segments. Aside from axons, we also found a posterior extension of the dendrites of Wave neurons in *DFz2* KD animals (*Figure 1—figure supplement 3*). However, the ectopic extension of dendrites was much shorter and occurred less frequently as compared to that of the axon. The conserved dendritic extension pattern suggests that a-Wave neurons still receive inputs from tactile sensory neurons in anterior segments in *DFz2* KD animals as in the wild type (*Takagi et al., 2017*). Thus, while it cannot be completely excluded that there are some changes in the inputs, the outputs are more likely to be rerouted by *DFz2* KD in a-Wave neurons.

To test if ectopic synapses are indeed formed in the extended Wave axon projections, we co-labeled wild-type and *DFz2* KD Wave neurons with a synaptic marker Bruchpilot (Brp; *Wagh et al., 2006*). We labeled A4-Wave by using MCFO and quantified the distribution of boutons (defined as axonal varicosities that have more than twice the width of axonal process) and whether they are co-labeled with Brp. While not all the boutons were Brp$^+$ (*Figure 2A and B*), we observed that synapses are formed at all the neuromere where the A4-Wave axon is projecting (*Figure 2C and D*). Importantly, the number of synapses was shifted toward posterior neuromeres upon *DFz2* KD, which is a pattern similar to the axon projection phenotype. The fraction of synapses over all boutons was relatively uniform across neuromeres (*Figure 2E*), indicating that presynaptic varicosities are a decent indicator of synapses. These observations suggest that *DFz2* KD results in ectopic downstream Wave connectivity, possibly altering the motor command upon its activation.

We also found that posterior axon extension of the most posterior p-Wave neuron (A6-Wave) is shortened in *DFz4* KD animals (*Figure 3A–I*). This observation suggests that *DFz4* mediates attraction of the p-Wave axons toward the posterior end. Neither posterior extension of other Wave axons (*Figure 3C–F*) nor the anterior extension of A6 or other Wave axons (*Figure 3C–G*) was affected, implying that DFz4 promotes axon extension of Wave axons selectively in A6-Wave and toward the posterior end (summarized in *Figure 3J*). Also, no abnormality was observed in the dendrites of a- or p-Wave neurons (data not shown). *DFz4* overexpression did not yield any obvious morphological changes (*Figure 3H and I*), suggesting that DFz4 expression alone is not sufficient to change the axon extension of Wave neurons.

## DWnt4 is a graded cue that regulates A-P Wave axon guidance

We next sought to identify the ligand(s) of DFz2/DFz4 receptors that mediate Wave axon guidance. We first examined the overall extension pattern of Wave neurites in four *Drosophila Wnt* mutants

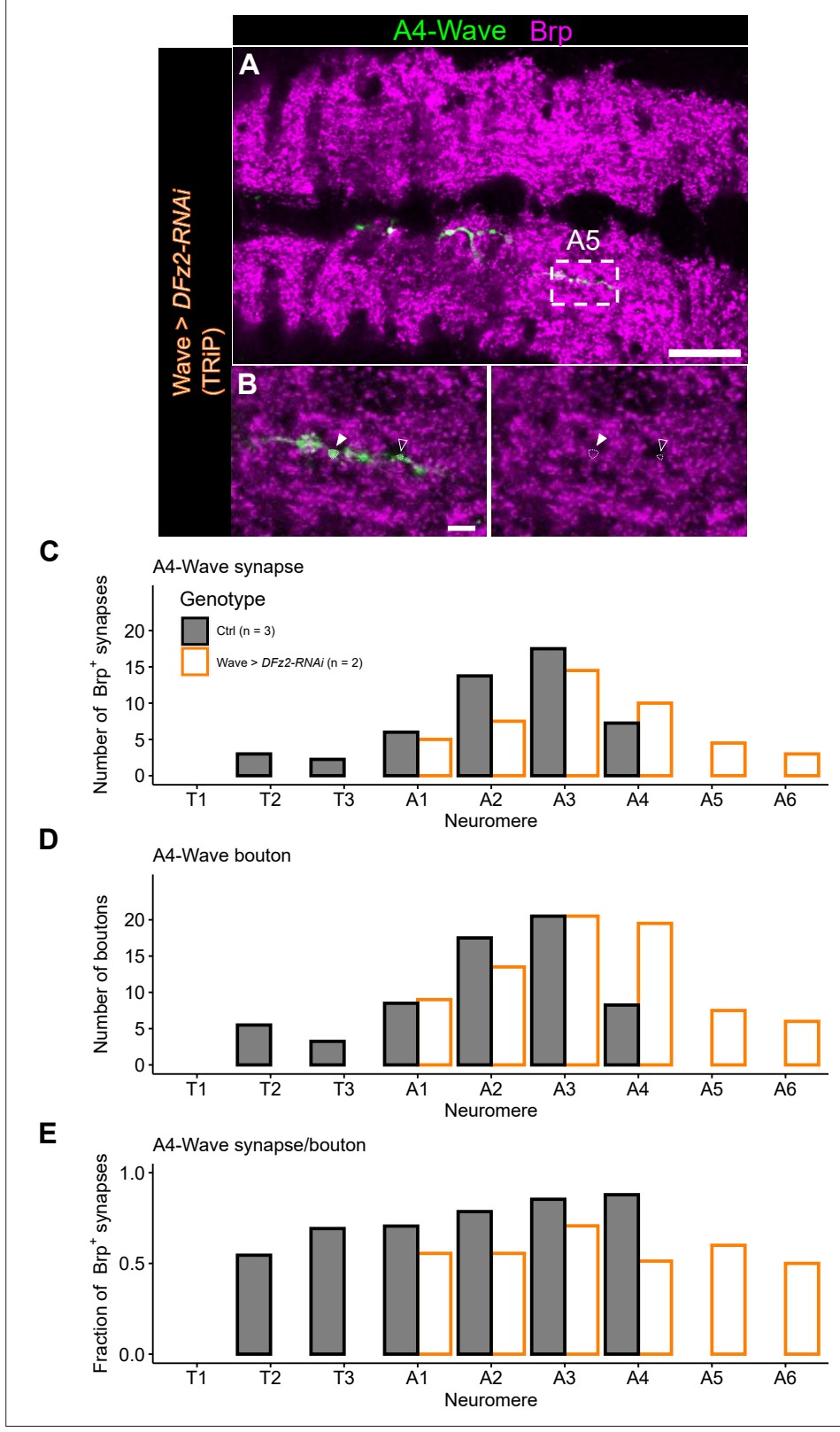

**Figure 2.** Ectopic synapses are formed in extended Wave axon projection. (**A**) Dorsal view of a ventral nerve cord containing a single labeled A4-Wave (green) and stained with a presynaptic marker Bruchpilot (Brp) (magenta). Scale bar = 50 μm. (**B**) Enlarged images of neuromere A5. White and black arrowheads indicate Brp$^+$ and Brp$^-$ boutons, respectively. Scale bar = 5 μm. (**C**) Number of A4-Wave synapses (Brp$^+$ boutons) in each neuromere.

*Figure 2 continued on next page*

*Figure 2 continued*

(**D**) Number of all A4-Wave boutons in each neuromere. (**E**) Fraction of A4-Wave synapses over all boutons in each neuromere.

---

($DWnt4^{C1/EMS23}$, $DWnt5^{400}$, $DWnt6^{KO}$, and $DWnt8^{KO1}$) and found that p-Wave axon extension toward the posterior end was shortened in *DWnt4* mutants, which is reminiscent of *DFz4* KD animals (***Figure 1— figure supplement 2E***; ***Figure 4A–C***). DWnt4 is known to have binding affinity to both DFz2 and DFz4 (***Wu and Nusse, 2002***), making it a strong candidate as a shared ligand of these receptors in Wave axon guidance. Further analyses with single-Wave labeling (by using heat-shock FlpOut) also revealed abnormalities in a-Wave; posterior extension of a-Wave axons was elongated, as was observed in *DFz2* KD animals (***Figure 4D, E, and G***). Additionally, a-Wave axon extension toward the anterior end was shortened in *DWnt4* mutants (***Figure 4D, E, and F***), which was not observed in DFz2 or DFz4 receptor KD animals, suggesting that DWnt4 regulates anterior extension of a-Wave via other receptor(s). The fact that *DWnt4* mutants recapitulated both phenotypes observed in *DFz2* and *DFz4* KD animals (***Figure 4H***) suggests that both DFz2 and DFz4 use DWnt4 as a guidance cue to regulate the extension of Wave axons along the A-P axis but with opposite 'valences'.

Polarized extension of axons along the A-P axis could be regulated by gradients of the guidance cues and/or their receptors (***Kirszenblat et al., 2011***; ***Lyuksyutova et al., 2003***). We therefore asked if there is any difference in the expression of DWnt4 and DFz2 along the A-P axis in the VNC. It has been previously suggested that *DWnt4* mRNAs are strongly expressed at the posterior end of the VNC (***Hessinger et al., 2017***). Our analyses of *DWnt4* knock-in GAL4 line ($Wnt4^{MI03717-Trojan-GAL4}$) further confirmed that the expression of *DWnt4* shows a gradient with higher expression toward the posterior end of the VNC (***Figure 5A and B***; ***Figure 5—figure supplement 1***). Furthermore, DWnt4 immunostaining revealed a similar gradient within the neuropil (***Figure 5C and D***; ***Figure 5—figure supplement 2A and B***), indicating that protein expression is also polarized. These observations support the notion that DWnt4 regulates axon guidance along the A-P axis by forming a gradient along the axis.

Next, we performed DFz2 immunostaining to investigate the receptor expression along the A-P axis. We observed a weak bidirectional gradient of DFz2 expression along the A-P axis, with a slightly higher level of expression in the anterior part of the VNC neuropil (***Figure 5E and F***; ***Figure 5—figure supplement 2C and D***), as well as in the posterior end (***Figure 5—figure supplement 3***). Collectively, these results are consistent with the idea that complementary expression of DWnt4 and DFz2 regulates Wave axon extension via repulsion: Wave axons with higher levels of DFz2 and thus with higher sensitivity to repulsive DWnt4 signaling extend to the anterior VNC where levels of DWnt4 are low (***Figure 5G***). Due to the lack of reagents that allow visualization of DFz4 expression, whether DFz4 also shows graded expression along the A-P axis remains to be determined. Nonetheless, based on the phenotype of *DFz4* KD (***Figure 3***) and the expression of DWnt4, it is likely that DFz4 guides p-Wave axons posteriorly by recognizing DWnt4 as an attractive cue (***Figure 5G***).

## Motor commands of a-Wave neurons are altered by *DFz2* KD

Having characterized the molecular mechanisms involved in morphological divergence of Wave neurons, we asked if the behavioral outputs of Wave are altered upon such anatomical perturbation. We previously used FLP-Out optogenetics experiments to show that activation of single a- and p-Wave neurons in freely behaving larvae induces backward and fast-forward locomotion, respectively, and co-activation of both a- and p-Wave neurons induces rolling (***Takagi et al., 2017***). Consistent with this, optogenetic stimulation of all Wave neurons in the larvae induces a mixture of fast-forward locomotion, backward locomotion, and roll/bend (***Takagi et al., 2017***; ***Figure 6A***). Fast-forward locomotion occurs in the context of escape from nociceptive inputs and is considered a distinct behavior from normal forward locomotion (e.g. during navigation) of the larvae (***Ohyama et al., 2013***; ***Ohyama et al., 2015***). Since it is technically challenging to perform FLP-Out optogenetics in RNAi KD animals, we tested the impact of *DFz2* KD and accompanied alteration in axon extension on Wave neuron's motor outputs by activating all Wave neurons in the larvae. In control animals, as observed previously (***Takagi et al., 2017***), the stimulation induced fast-forward locomotion (***Figure 6B***), backward locomotion (***Figure 6B'***), and rolling (***Figure 6B"***). By contrast, in *DFz2* KD animals, the stimulation induced fast-forward locomotion (***Figure 6C***) and rolling (***Figure 6C"***) but not backward locomotion (***Figure 6C'***). Importantly, we found that the speed of the evoked fast-forward locomotion (during

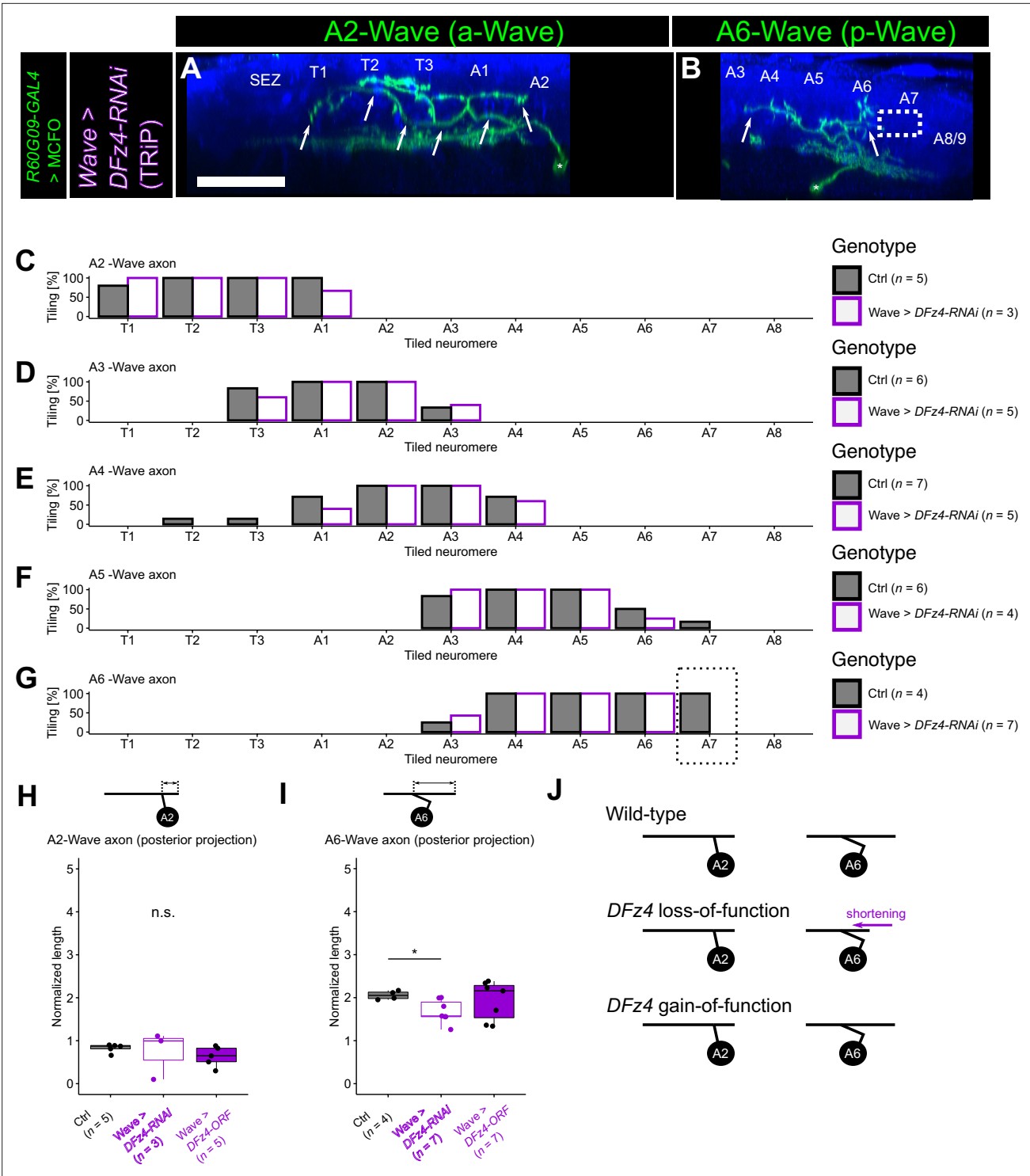

**Figure 3.** DFz4 specifically promotes axon extension toward the posterior end in A6-Wave. (**A, B**) *DFz4* knockdown (KD) A2 (**A**) and A6 (**B**) Wave neuron using the TRiP RNAi line. Note the shortening of the A6 axon toward the posterior end (dotted box in **B**, see *Figure 1B and C* for control). Scale bar = 50 μm. (**C–G**) Tiling profile of A2 to A6 Wave axons in control and *DFz4* knocked-down animals. *n* indicates the number of neurons. Tiling percentage indicates the fraction of samples (single Wave neurons) whose Wave axon innervated the corresponding neuromere. The same analyses as in *Figure 1F–J* but for *DFz4* knocked-down animals. The control data in *Figure 1F–J* are shown as references. (**H, I**) Quantification of axon extension toward the posterior end in A2 (**H**) and A6 Wave (**I**) measured in MultiColor FlpOut (MCFO) images, following KD and overexpression of *DFz4*, respectively. *: p<0.05, Welch's t-test with Bonferroni correction. The control data shown in *Figure 1K and L* are reused for this analysis. (**J**) Summary of *DFz4* KD and overexpression phenotypes.

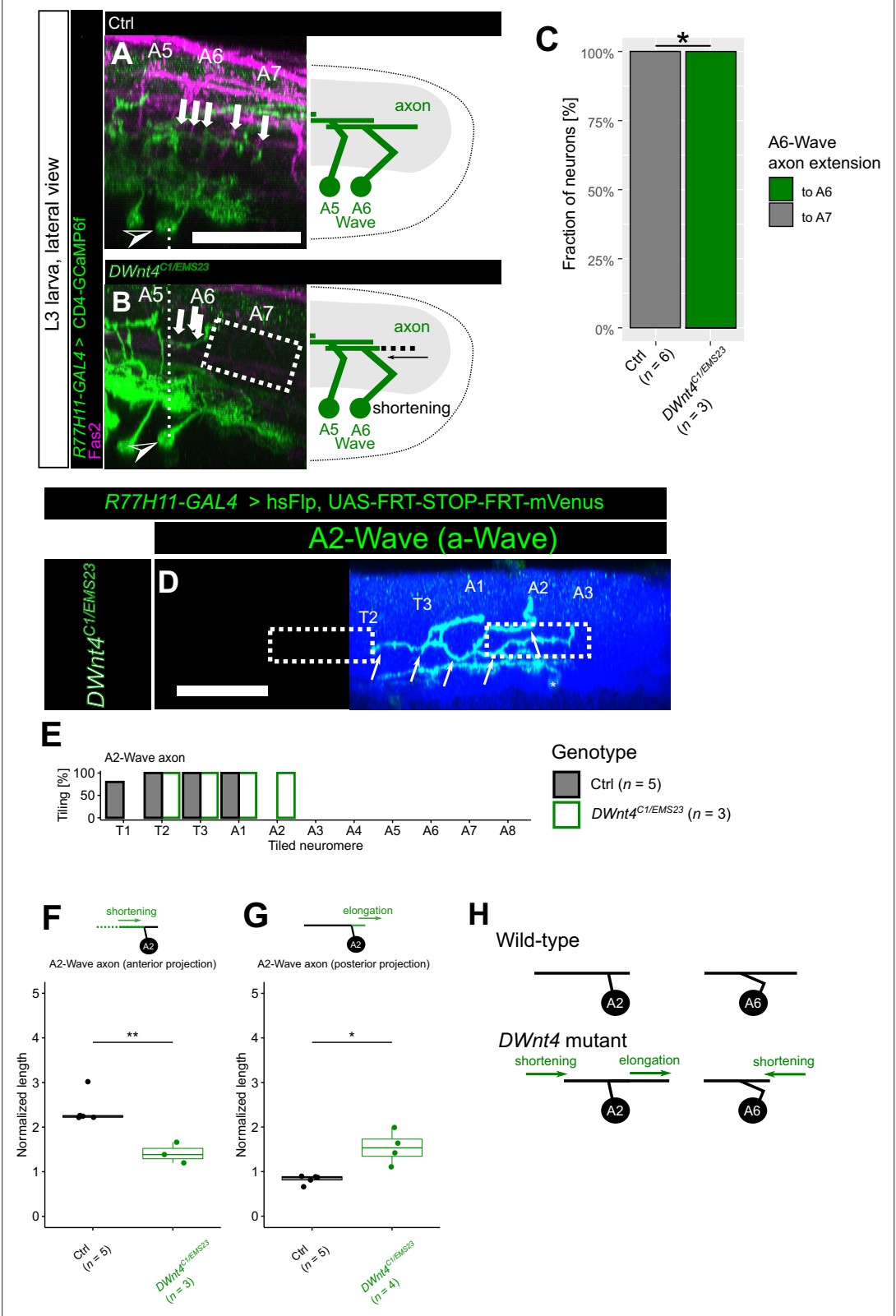

**Figure 4.** DWnt4 regulates anterior-posterior (A-P) extension of Wave axons. (**A–C**) DWnt4 promotes posterior axon extension of p-Wave. (**A, B**) Examples of axon morphologies toward the posterior end (white arrows) in control (**A**) and *DWnt4*<sup>C1/EMS23</sup> mutant (**B**) animals. The posterior end of the axon, derived from p-Wave, is shortened in *DWnt4* mutant. Dotted boxes indicate the abnormal shortening of the axon. (**C**) Quantification of A6-Wave axon extension to segment A6 or A7. n=6 (control) and 3 (*DWnt4*<sup>C1/EMS23</sup>) neurons, respectively: p=0.0119, Fisher's exact test. (**D–G**) DWnt4 regulates

*Figure 4 continued on next page*

*Figure 4 continued*

axon extension of a-Wave. (**D**) A2-Wave in *DWnt4$^{C1/EMS23}$* is visualized using heat-shock FlpOut. Note the elongation of its axon toward the posterior end, as well as the shortening of its axon toward the anterior end (dotted boxes). Scale bar = 50 µm. See *Figure 1B* for control. (**E**) Tiling profile of A2-Wave axons in control and *DWnt4$^{C1/EMS23}$* mutant animals. *n* indicates the number of neurons. Tiling percentage indicates the fraction of samples (single Wave neurons) whose Wave axon innervated the corresponding neuromere. The control data in *Figure 1F* are reused and shown as references for different comparisons. (**F, G**) Quantification of the relative length (see Materials and methods details) of the anterior and posterior fraction of A2-Wave axons between control (visualized by MultiColor FlpOut [MCFO]) and mutant (visualized by heat-shock FlpOut) animals. (**F**) Quantification of the anterior fraction of A2-Wave axons. (**G**) Quantification of the posterior fraction of A2-Wave axons. *n* indicates the number of neurons. *: p<0.05, **: p<0.01, Welch's t-test. The control data are replotted from *Figure 1K* for different comparisons. (**H**) Summary of *DWnt4* phenotypes.

stimulation) was faster in *DFz2* KD animals, while the baseline speed was unchanged (*Figure 6D and E*), indicating that fast-forward locomotion is enhanced in *DFz2* KD animals upon Wave activation. In summary, *DFz2* KD in Wave neurons augments their motor command of fast-forward locomotion while diminishing that of backward locomotion.

## Altered behavioral response to tactile stimuli in *DFz2* KD larvae

The differential motor behaviors commanded by Wave neurons are triggered by tactile stimuli at different body parts (*Figure 1A and A'*; *Takagi et al., 2017*). Does the altered axon projection of Wave neurons in *DFz2* KD animals impact the behavioral response evoked by tactile stimuli? Specifically, does selective a-Wave activation by natural tactile stimuli in the head induce fast-forward instead of backward locomotion, as observed in the optogenetic experiment (*Figure 6*)? We tested these possibilities by presenting freely behaving larvae with a gentle touch on their head using a von Frey filament, which activates a- but not p-Wave neurons (*Takagi et al., 2017*). The gentle touch responses are known to be stereotypic but range from weak to strong: continuation of forward crawls (score 0), halting followed by crawl resumption (score 1), turning followed by crawl resumption (score 2), backing up once (score 3), and backing up multiple times (score 4) in *Drosophila melanogaster* larvae (*Kernan et al., 1994*, *Figure 7A*). We found that *DFz2* KD by using two different RNAi lines in Wave neurons reduced backward locomotion responses (scores 3 and 4) while increasing turning responses (score 2) to head touch (*Figure 7B, B', C, and C'*). Since turning is not induced by Wave activation (*Takagi et al., 2017*), it is likely that the increased turning is due to the disinhibition of turning-inducing pathways (through other second-order tactile interneurons) by the deterioration of the backward-inducing pathway (through Wave). Critically, the speed of forward locomotion after each touch (usually after turning) was faster in *DFz2* KD animals (*Figure 7B" and C"*), implying that fast-forward locomotion was induced in response to tactile stimulus. Taken together, our results suggest that changes in the axon guidance of Wave neurons alter the behavioral strategy of the larvae in response to external stimuli by altering the neurons' command from backward to fast-forward locomotion (*Figure 7D*).

## Discussion

To understand the circuit mechanisms underlying behavior, perturbation of neurons and circuits while observing the behavior has been a successful paradigm (*Luo et al., 2018*). Most studies linking change in nervous system perturbation and behavioral alteration are either changes in physiology and/or mainly in the sensory periphery (*Ardesch et al., 2019*; *Kim et al., 2017*; *Seeholzer et al., 2018*; *Tinbergen, 1951*). Here, we revealed a link between axon guidance and circuit function by identifying molecular mechanisms underlying the divergent axon extension of segmentally homologous Wave neurons and showing changes in the axon guidance alter motor commands (*Figure 7D*). The results illustrate several crucial features on how neural circuits mediating adaptive sensorimotor transformation form during development, and on how the evolution of the nervous system might lead to the acquisition of new behaviors.

## Universal roles of Wnt/Fz signaling in A-P neurite guidance

Secreted proteins Wnts and their receptors Frizzled are evolutionarily conserved families of proteins. Wnt/Fz signaling is known to regulate A-P axon pathfinding in the nervous system of mammals and *Caenorhabditis elegans* (*Hilliard and Bargmann, 2006*; *Lyuksyutova et al., 2003*; *Salinas and Zou, 2008*). For instance, commissural neurons in the mouse spinal cord after crossing the midline extend

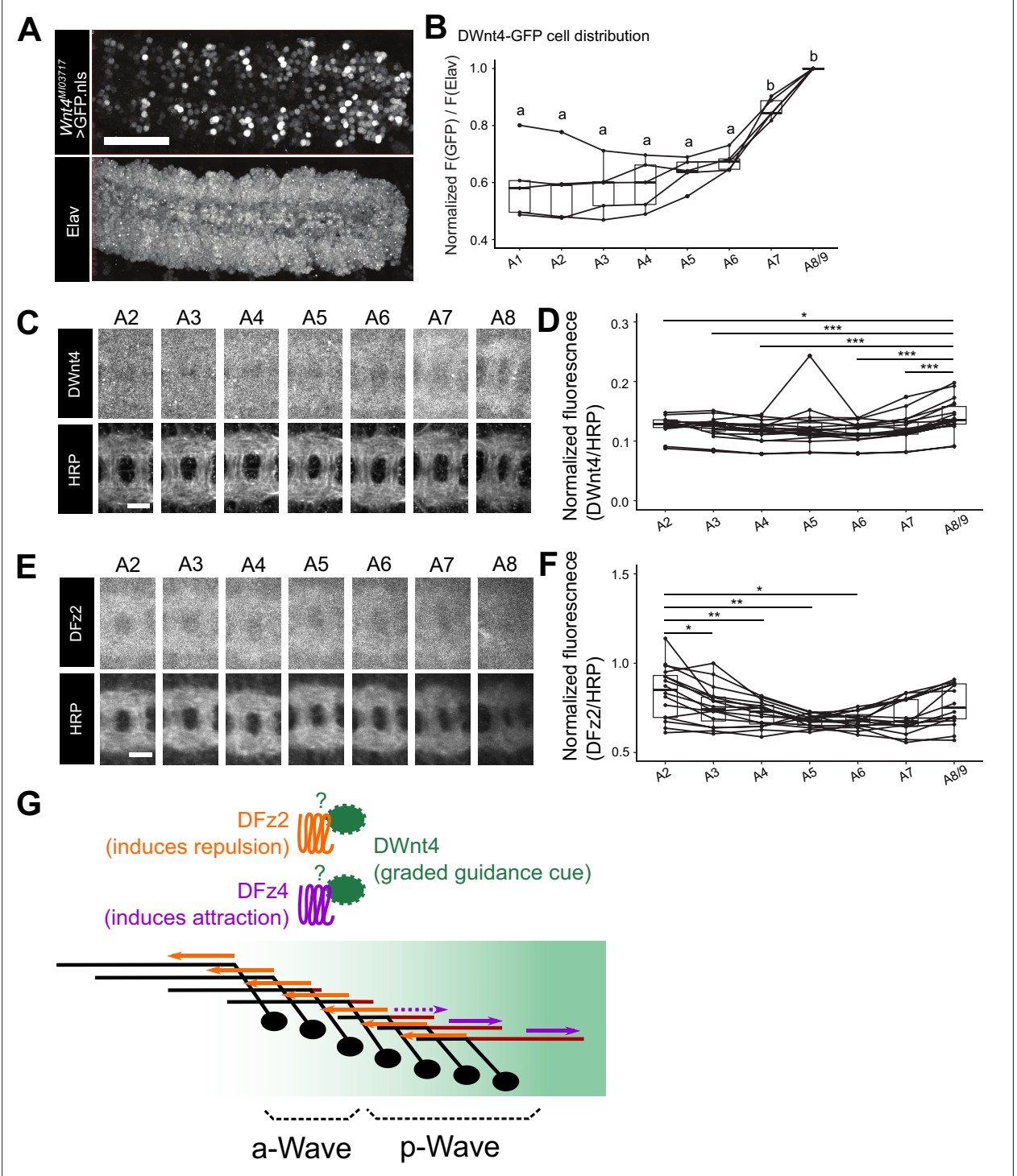

**Figure 5.** Complementary graded expression of DWnt4 and DFz2 along the anterior-posterior (A-P) axis. (**A, B**) DWnt4-GFP cells in the embryonic CNS. (**A**) Distribution of DWnt4-GFP cells. Nuclear-localized GFP is expressed under the regulation of *Wnt4^MI03717-Trojan-GAL4*. anti-Elav counterstaining was performed to label all neurons and normalize the GFP signal. GFP expression is stronger in posterior than anterior ventral nerve cord (VNC). Scale bar = 50 μm. (**B**) Quantification of normalized GFP signals with respect to Elav signals in each neuromere (where 1 denotes the maximum). *n*=5 animals. Not having a common alphabet indicated above the plots between groups indicates statistical significance (α=0.05, Tukey's HSD test). (**C, D**) Expression of DWnt4 protein in the neuropil of each neuromere. (**C**) DWnt4 and HRP immunostaining. Scale bar = 10 μm. (**D**) Normalized DWnt4/HRP signals in each neuromere are calculated. *n*=18 samples from 9 animals *: p<0.05, ***: p<0.001, paired t-test (comparison to A8). (**E, F**) Expression of DFz2 protein in the neuropil of each neuromere. (**E**) DFz2 and HRP immunostaining. Scale bar = 10 μm. (**F**) Normalized DFz2/HRP signal in each neuromere. *n*=16 samples from 8 animals *: p<0.05, **: p<0.01, paired t-test (comparison to A2). (**G**) Model of segment-specific axon guidance in Wave neurons. DWnt4 serves

*Figure 5 continued on next page*

*Figure 5 continued*

as a graded axon guidance cue (concentrated toward the posterior end of VNC) and is recognized by DFz2/DFz4 receptors as repulsive/attractive cue, respectively. DFz2 functions in both a-Wave and p-Wave, whereas DFz4 functions selectively in p-Wave.

The online version of this article includes the following figure supplement(s) for figure 5:

**Figure supplement 1.** DWnt4-GFP gradient is retained in 3rd instar larvae.

**Figure supplement 2.** Expression of DWnt4 and DFz2 is suppressed in mutants.

**Figure supplement 3.** DFz2 gradient is bidirectional.

their axons anteriorly by recognizing an anterior-high gradient of Wnt4 via the Frizzled-3 receptor (*Lyuksyutova et al., 2003*). Similarly, lin-44/Wnt and lin-17/Frizzled regulate the A-P extension of the PLM mechanosensory neuron in *C. elegans* (*Hilliard and Bargmann, 2006*). Although Wnt/Fz proteins are known to regulate axon guidance in *Drosophila* (e.g. *Inaki et al., 2007*; *Sato et al., 2006*; *Yoshikawa et al., 2003*), whether they play roles in A-P neurite guidance in the nerve cord was not previously known. The present study demonstrates this in *Drosophila* and thus points to the universal roles played by Wnt/Fz signaling in A-P neurite guidance in the animal kingdom.

We have shown that Frizzled receptors DFz2 and DFz4 regulate A-P axon guidance of homologous Wave neurons in segment-specific manner and in opposite valences, presumably by using DWnt4 as a common external guidance cue. Our results suggest that DFz4 mediates attraction of posterior Wave neurons toward the source of DWnt4 at the posterior end, whereas DFz2 mediates repulsion of both anterior and posterior Wave neurons away from the DWnt4 source. The presence of two graded signaling systems with antagonistic effects may aid in establishing reliable A-P axon guidance as proposed by theoretical models (*Gierer, 1987*). Our functional analyses of DFz2 and DFz4 receptors support the idea that the two signaling systems cooperate to guide the projection of Wave axons along the A-P axis. While this study focused on Wave neurons, it is likely that Wnt/Fz signaling also regulates A-P guidance of many other longitudinal axons, since DFz2 immunoreactivity was seen on a large fraction of axons in the anterior VNC.

## Diversification of segment-specific motor commands by Fz signaling

The command neuron concept is a widely appreciated notion that a specific type of interneuron serves as a linking node between external inputs and appropriate motor outputs (*Bouvier et al., 2015*; *Kupfermann and Weiss, 1978*; *Wiersma and Ikeda, 1964*). For command neurons to function in adaptive manners, they must be precisely wired to appropriate upstream and downstream circuits (*Faumont et al., 2012*). Indeed, modulation of input connectivity to command neurons is known to alter sensory-evoked behaviors (*Tenedini et al., 2019*; *Valdes-Aleman et al., 2021*). However, whether changes in command neuron axon guidance alter their motor outputs remained unknown. More importantly, how homologous command neurons might diverge their function to encode adaptive behavior(s) has been a long-standing question.

Here, we found that *DFz2* KD appears to influence both the axon projection pattern of a-Wave neurons and their associated motor outputs, shifting the behavioral response from backward to forward locomotion. Upon optogenetic activation, *DFz2* KD Wave neurons triggered significantly less backward locomotion and instead promoted fast-forward locomotion (*Figure 6*). Furthermore, DF2 manipulation altered larval responses to head touch, favoring turning-and-running over backing up (*Figure 7*). It is interesting that these behavioral changes were induced even though DFz2 was specifically manipulated in Wave neurons and therefore other known secondary neurons downstream of touch sensors and nociceptors (*Eschbach and Zlatic, 2020*; *Hu et al., 2020*) were likely unaffected. These findings imply that the specificity of Wave-mediated motor commands plays important roles in shaping larval behavior. It should be noted, however, that while the DFz2 KD phenotypes are consistent with aberrant wiring of a-Wave axons to forward locomotion-promoting circuits, the contributions of dendritic alterations remain unclear. Additionally, altered perception and projection of p-Wave neurons may directly or indirectly contribute to the observed behavioral phenotypes, particularly in response to mechanical stimulation.

Segmental units in the spinal cords in vertebrates and VNC in invertebrates receive sensory inputs from corresponding body segment(s) and reroute the information to distinct motor outputs. The fact that *DFz2* KD a-Wave neurons function like p-Wave neurons and induce forward locomotion suggests

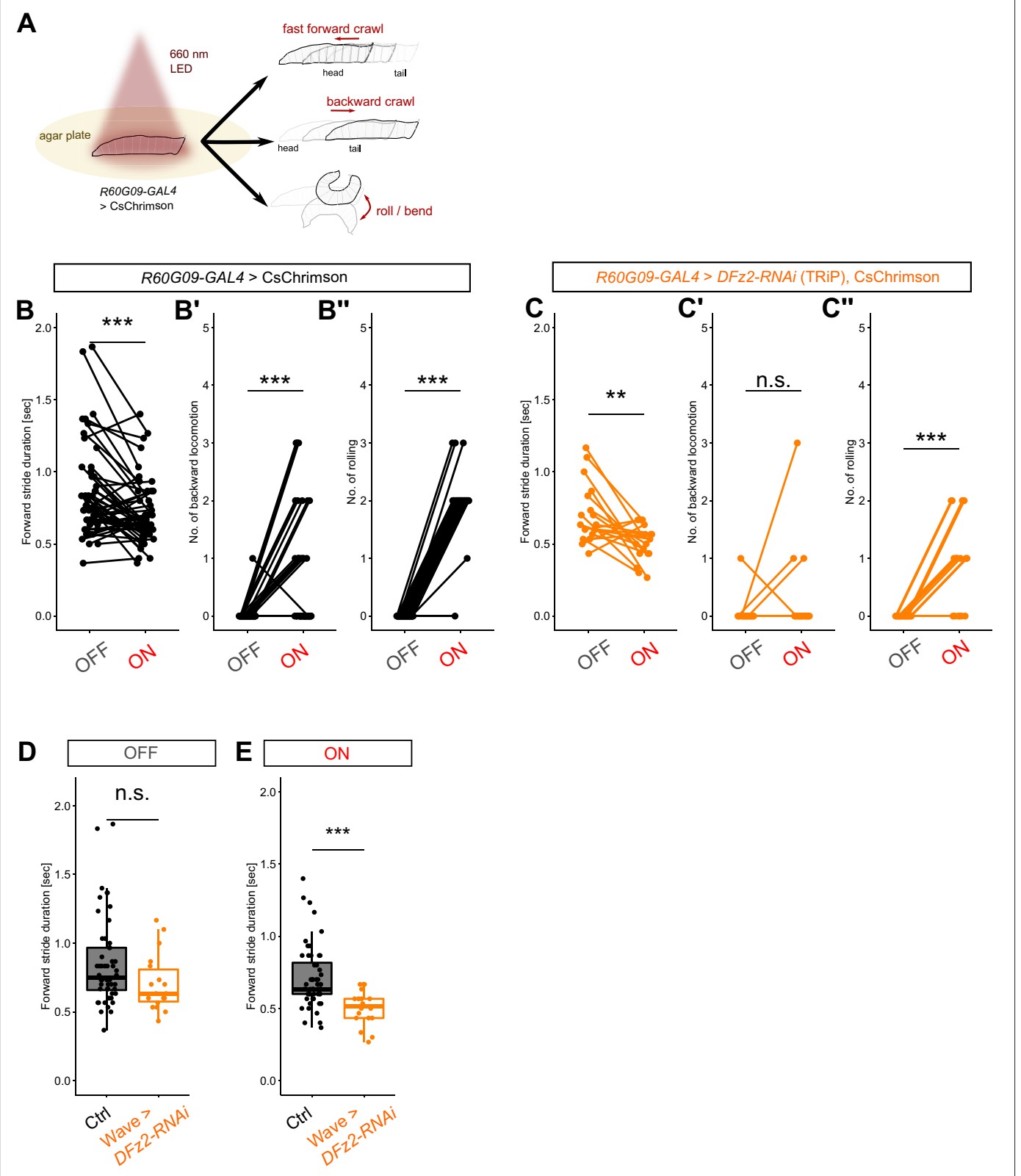

**Figure 6.** *DFz2* knockdown (KD) in Wave neurons alters motor commands in vivo. (**A**) Schematic of optogenetics assay in vivo. (**B–B''**) Comparison of the larval behavior between LED OFF and ON conditions in control animals. LED illumination decreases stride duration of forward locomotion (i.e. induces fast-forward locomotion) (**B**), induces backward locomotion (**B'**), and induces rolling (**B''**). *n*=52 animals. ***: p<0.001, Wilcoxon's signed-rank test. (**C–C''**) Comparison of the larval behavior between LED OFF and ON conditions in *DFz2* knocked-down [TRiP] animals. LED illumination decreases stride

*Figure 6 continued on next page*

*Figure 6 continued*

duration of forward locomotion (i.e. induces fast-forward locomotion) (**C**), does not induce backward locomotion (**C'**), and induces rolling (**C''**). n=18 animals. **: p<0.01, ***: p<0.001, Wilcoxon's signed-rank test. (**D, E**) Comparison of the stride duration of forward locomotion between control and *DFz2* knocked-down animals in LED OFF (**D**) and ON (**E**) conditions. The stride duration showed no significant difference in LED OFF period (**D**) but was shorter in *DFz2* knocked-down animals in LED ON period (**E**). Mean ± SD stride durations: 0.85±0.32 (Ctrl, OFF), 0.71±0.22 (Ctrl, ON), 0.71±0.21 (*DFz2* KD, OFF), 0.50±0.12 (DFz2 KD, ON). ***: p<0.001, Mann-Whitney's U-test. Note that the stride durations of the control and *DFz2* KD animals are slightly different in the OFF condition, although this is not statistically significant. In addition, the effect size of Wave activation on mean stride duration is –0.14 (s) in control while –0.21 (s) in *DFz2* KD, which we interpret as *DFz2* KD resulting in stronger fast-forward locomotion upon Wave activation.

that posterior axon projection and connection with forward-inducing circuits are a default state of Wave neurons and that suppression of posterior extension by Wnt/DFz2 is required for a-Wave to connect to alternative downstream circuits (those inducing backward locomotion) and acquire an alternative motor command. Our findings suggest a possible mechanism by which command functions of segmentally homologous neurons may diverge during development, potentially contributing to diverse sensory-motor transformations.

## Flexible modulation of motor commands by A-P axon guidance

Since the way in which the nervous system carries out sensory-to-motor transformation is different between animal species in order to meet their respective ecological niches, the connectivity must be able to change flexibly during the evolution of the nervous system. If and how such connectivity changes occur during evolution remains poorly understood (*Katz, 2011*; *Katz, 2016*; *Tosches, 2017*). While a few studies demonstrated that the evolution of nervous system functions is indeed accompanied by connectivity changes (*Ardesch et al., 2019*; *Sakurai et al., 2011*; *Seeholzer et al., 2018*), the underlying developmental mechanisms remain unknown. Since command neurons link specific sensory inputs to appropriate motor outputs, changes in their wiring are likely to be a powerful mechanism by which animals acquire new sensory-evoked behavioral strategies that match changing environments without perturbing the overall network stability. In this study, we propose a simple developmental mechanism that may contribute to such an evolutionary process. By rerouting the axon of a single command neuron (a-Wave) to an ectopic location (posterior neuromeres), we observed changes in larval behavior. These findings imply the existence of flexible neural mechanisms that could support a 'plug-and-play' style modulation of motor commands as follows.

First, simply guiding the axon to a novel position appears to be sufficient for synapse formation with alternative downstream circuits. Since a-Wave neurons acquired the ability to induce fast-forward locomotion in *DFz2* KD animals, these neurons likely formed synapses with neural circuits that trigger the behavior, presumably the downstream targets of p-Wave. It is known that neurons have robust capacity to form synapses with nontarget cells, for instance, when placed in an ectopic position or when normal targets are deleted (*Bekkers and Stevens, 1991*; *Cash et al., 1992*; *Hassan and Hiesinger, 2015*; *Shen and Bargmann, 2003*; *Xu et al., 2019*). Furthermore, since a-Wave and p-Wave neurons are segmental homologs and likely share a large fraction of gene expression including those involved in synapse formation, a-Wave neurons once placed in the target region of p-Wave may readily form synapses with the p-Wave targets.

Second, there appears to be mechanism(s) that suppress functional connectivity to alternative downstream circuits so that only one behavioral output is induced. While a-Wave neurons in *DFz2* KD animals form ectopic axon extension posteriorly, the original anterior axon extension, which normally connects to backward-inducing circuits, remains largely intact morphologically. Nonetheless, the ability of a-Wave to induce backward locomotion is greatly reduced, and instead, forward-inducing ability is conferred. This suggests that connectivity from a-Wave axons to backward-inducing circuits is functionally suppressed, possibly by weakening of functional synapses between the synaptic partners and/or via reciprocal inhibition between the antagonistic downstream circuits.

The observation that switching in motor commands can be induced by manipulating a single gene in a single class of neurons raises the possibility that the Wave circuitry may be relatively amenable to evolutionary modification (*Kirschner and Gerhart, 1998*), potentially allowing adaptation to species-specific tactile environments. Supporting this idea, we observed that larval responses to head touch, where Wave neurons are thought to play major roles, differ among closely related species of *Drosophila* (S Takagi and AN, unpublished data). These results point to a degree of flexibility in the

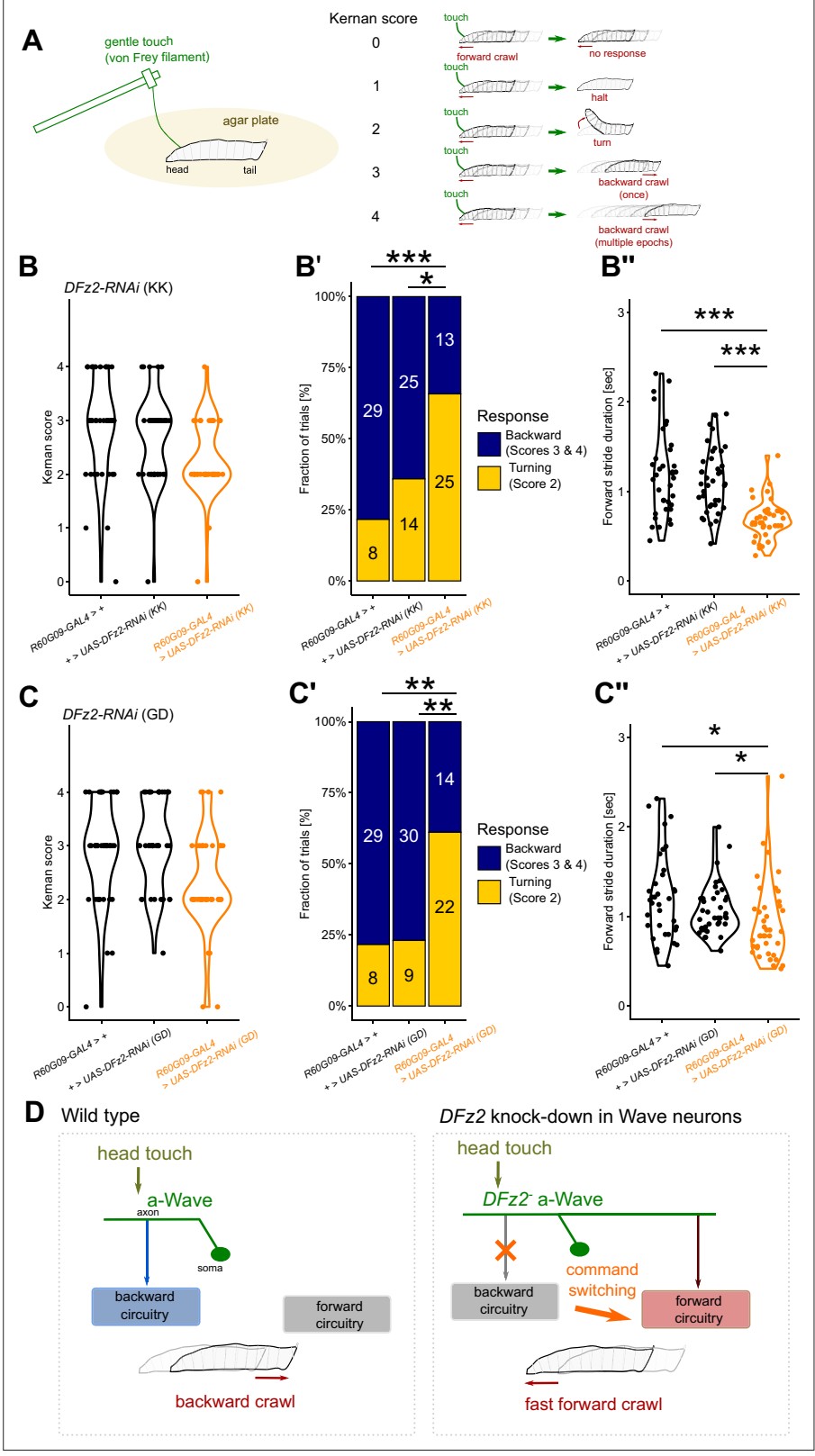

**Figure 7.** Modulation of tactile-evoked behavior by *DFz2* knockdown (KD) in Wave neurons. (**A**) Left panel: Scheme of gentle-touch assay using von Frey filament. Right panel: Stereotypic behavioral responses induced by the gentle head touch, categorized according to the Kernan score (**Kernan et al., 1994**). (**B–C"**) Alteration in behavior responses upon *DFz2* KD using UAS-*DFz2-RNAi* [KK] (**B–B"**) or UAS-*DFz2-RNAi* [GD] (**C–C"**). (**B, C**) Distribution of

*Figure 7 continued on next page*

*Figure 7 continued*

the Kernan score. Driver control; *R60G09-GAL4/+*, effector control; UAS-*DFz2-RNAi/+*, and experimental group (*R60G09-GAL4>UAS-DFz2-RNAi*). *n*=40 trials each. (**B′, C′**) Fraction of behavioral responses classified as backward (scores 3 and 4) and turning (score 2). The numbers indicate the trials that induced the classified response. *: p<0.05, **: p<0.01, ***: p<0.001, Fisher's exact test with Bonferroni correction. (**B′′, C′′**) Quantifications of stride duration of forward locomotion following gentle touch. Note that smaller stride duration indicates faster forward locomotion. *n*=36 (driver control), 40 (effector control), 39 (**B′′**), and 40 (**C′′**) (experimental) trials. *: p<0.05, ***: p<0.001, Steel-Dwass test. (**D**) Summary of the current study.

command system that might facilitate behavioral evolution. Accordingly, the present study not only provides insight into how segment-specific motor commands may be established along the A-P axis during development, but also highlights a potential molecular mechanism that could underlie evolutionary changes in the command system. Since Wnt/Fz-mediated regulation of A-P axon guidance is conserved across phyla, our finding may reflect a broader principle relevant to the formation and diversification of the command system in the nerve cord.

# Materials and methods

**Key resources table**

| Reagent type (species) or resource | Designation | Source or reference | Identifiers | Additional information |
|---|---|---|---|---|
| Strain, strain background (*Drosophila melanogaster*) | yw | Bloomington Drosophila Stock Center | RRID:BDSC_6598 | |
| Strain, strain background (*D. melanogaster*) | DWnt4<sup>C1</sup> | Bloomington Drosophila Stock Center; *Cohen et al., 2002* | RRID:BDSC_6651 | Null allele of DWnt4 |
| Strain, strain background (*D. melanogaster*) | DWnt4<sup>EMS23</sup> | KYOTO Drosophila Stock Center; *Cohen et al., 2002* | KYOTO:108–974 | Null allele of DWnt4 |
| Strain, strain background (*D. melanogaster*) | DWnt5<sup>400</sup> | Bloomington Drosophila Stock Center; *Fradkin et al., 2004* | RRID:BDSC_64300 | Null allele of DWnt5 |
| Strain, strain background (*D. melanogaster*) | DWnt6<sup>KO</sup> | Bloomington Drosophila Stock Center; *Doumpas et al., 2013* | RRID:BDSC_76311 | Deletion of exon 1 of DWnt6 |
| Strain, strain background (*D. melanogaster*) | DWnt8<sup>KO1</sup> | Bloomington Drosophila Stock Center; *Gordon et al., 2005* | RRID:BDSC_38407 | Knockout allele of DWnt8 (aka wntD) |
| Strain, strain background (*D. melanogaster*) | UAS-CD4-tdGFP | Bloomington Drosophila Stock Center | RRID:BDSC_35836 | A membrane-fused GFP |
| Strain, strain background (*D. melanogaster*) | UAS-CD4::GCaMP6f | *Takagi et al., 2017* | | A membrane-fused GCaMP |
| Strain, strain background (*D. melanogaster*) | MCFO-6 | Bloomington Drosophila Stock Center; *Nern et al., 2015* | RRID:BDSC_64090 | MCFO-6 |
| Strain, strain background (*D. melanogaster*) | 20XUAS >dsFRT >-CsChrimson::mVenus, pBPhsFlp2::Pest | *Takagi et al., 2017* | | FLP-Out optogenetics construct |
| Strain, strain background (*D. melanogaster*) | MB120B-spGAL4 | *Takagi et al., 2017* | | GAL4.AD and GAL4.DBD combination specific to Wave neurons |
| Strain, strain background (*D. melanogaster*) | R60G09-GAL4 | Bloomington Drosophila Stock Center | RRID:BDSC_46441 | GAL4 line targeting Wave neurons |
| Strain, strain background (*D. melanogaster*) | R77H11-GAL4 | Bloomington Drosophila Stock Center; *Masson et al., 2020* | RRID:BDSC_39983 | Targets Wave neurons |
| Strain, strain background (*D. melanogaster*) | R77H11-LexA | Bloomington Drosophila Stock Center | RRID:BDSC_54720 | LexA version of R77H11 enhancer |
| Strain, strain background (*D. melanogaster*) | Wnt4<sup>MI03717-Trojan-GAL4</sup> | Bloomington Drosophila Stock Center | RRID:BDSC_67449 | GAL4 insertion near Wnt4 locus |

*Continued on next page*

*Continued*

| Reagent type (species) or resource | Designation | Source or reference | Identifiers | Additional information |
|---|---|---|---|---|
| Strain, strain background (*D. melanogaster*) | elav-GAL4[3E1] | *Davis et al., 1997* | | Targets all neurons |
| Strain, strain background (*D. melanogaster*) | UAS-mCherry.VALIUM10 | Bloomington Drosophila Stock Center | RRID:BDSC_35787 | Control stock for RNAi reporter lines |
| Strain, strain background (*D. melanogaster*) | UAS-DFz-RNAi | Bloomington Drosophila Stock Center | RRID:BDSC_34321 | RNAi to knock down DFz |
| Strain, strain background (*D. melanogaster*) | UAS-DFz2-RNAi | Bloomington Drosophila Stock Center | RRID:BDSC_67863 | RNAi to knock down DFz2 |
| Strain, strain background (*D. melanogaster*) | UAS-DFz3-RNAi | Bloomington Drosophila Stock Center | RRID:BDSC_44468 | RNAi to knock down DFz3 |
| Strain, strain background (*D. melanogaster*) | UAS-DFz4-RNAi | Bloomington Drosophila Stock Center | RRID:BDSC_64990 | RNAi to knock down DFz4 |
| Strain, strain background (*D. melanogaster*) | UAS-drl-RNAi | Bloomington Drosophila Stock Center | RRID:BDSC_39002 | RNAi to knock down Drl |
| Strain, strain background (*D. melanogaster*) | UAS-Drl-2-RNAi [TRiP] | Bloomington Drosophila Stock Center | RRID:BDSC_55893 | RNAi to knock down Drl-2 |
| Strain, strain background (*D. melanogaster*) | UAS-Drl-2-RNAi [KK] | Vienna Drosophila Resource Center | VDRC:102192 | RNAi to knock down Drl-2 |
| Strain, strain background (*D. melanogaster*) | UAS-smo-RNAi | Bloomington Drosophila Stock Center | RRID:BDSC_43134 | RNAi to knock down Smo |
| Strain, strain background (*D. melanogaster*) | UAS-Corin-RNAi | Bloomington Drosophila Stock Center | RRID:BDSC_41721 | RNAi to knock down Corin |
| Strain, strain background (*D. melanogaster*) | UAS-DFz2-RNAi [GD] | Vienna Drosophila Resource Center | VDRC:44390 | RNAi to knock down DFz2 |
| Strain, strain background (*D. melanogaster*) | UAS-DFz2-RNAi [KK] | Vienna Drosophila Resource Center | VDRC:108998 | RNAi to knock down DFz2 |
| Strain, strain background (*D. melanogaster*) | UAS-DFz4-RNAi [KK] | Vienna Drosophila Resource Center | VDRC:102339 | RNAi to knock down DFz4 |
| Strain, strain background (*D. melanogaster*) | UAS-DFz2-ORF | FlyORF | FlyORF:F001187 | ORF line for DFz2 overexpression |
| Strain, strain background (*D. melanogaster*) | UAS-DFz4-ORF | FlyORF | FlyORF:F001662 | ORF line for DFz4 overexpression |
| Strain, strain background (*D. melanogaster*) | LexAop-RCaMP2 | KYOTO Drosophila Stock Center | KYOTO:118796 | *Takagi et al., 2017* |
| Antibody | Rabbit polyclonal anti-GFP | Frontier Institute | Af2020; RRID:AB_10615238 | 1/1000 |
| Antibody | Mouse monoclonal anti-Fas2 (1D4) | Developmental Studies Hybridoma Bank | RRID:AB_528235 | 1/10 |
| Antibody | Rat monoclonal anti-Elav (7E8A10) | Developmental Studies Hybridoma Bank | RRID:AB_528218 | 1/10 |
| Antibody | Mouse monoclonal anti-Repo (8D12) | Developmental Studies Hybridoma Bank | RRID:AB_528448 | 1/5 |
| Antibody | Mouse monoclonal anti-Brp (nc82) | Developmental Studies Hybridoma Bank | RRID:AB_2314866 | 1/100 |
| Antibody | Guinea pig polyclonal anti-GFP | Frontier Institute | Af1180; RRID:AB_2571575 | 1/1000 |
| Antibody | Rabbit monoclonal anti-HA (C29F4) | Cell Signaling Technology | RRID:AB_1549585 | 1/1000 |

*Continued on next page*

*Continued*

| Reagent type (species) or resource | Designation | Source or reference | Identifiers | Additional information |
|---|---|---|---|---|
| Antibody | Mouse monoclonal anti-V5 | Invitrogen | R960-25; RRID:AB_2556564 | 1/500 |
| Antibody | Rat monoclonal anti-FLAG | Novus Biologicals | NBP1-06712; RRID:1625981 | 1/500 |
| Antibody | Rabbit polyclonal anti-DWnt4 | *Cohen et al., 2002*; Gift from M Sato | | 1/100 |
| Antibody | Rabbit anti-DFz2 | *Packard et al., 2002*; Gift from G Alegre, V Budnik, and T Thomson | | 1/100 |
| Antibody | Goat polyclonal anti-rabbit IgG Alexa Fluor 488 | Invitrogen | A-11034; RRID:AB_2576217 | 1/300 |
| Antibody | Goat polyclonal anti-rabbit IgG Cy3 | Invitrogen | A-10520; RRID:AB_10563288 | 1/300 |
| Antibody | Goat polyclonal anti-mouse IgG Alexa Fluor 555 | Invitrogen | A-21424; RRID:AB_141780 | 1/300 |
| Antibody | Goat polyclonal anti-mouse IgG Cy5 | Invitrogen | A-10524; RRID:AB_2534033 | 1/300 |
| Antibody | Goat polyclonal anti-rat IgG Alexa Fluor 488 | Invitrogen | A-11006; RRID:AB_141373 | 1/300 |
| Antibody | Goat polyclonal anti-HRP Cy5 | Jackson ImmunoResearch | 123-175-021 | 1/300 |
| Antibody | Goat polyclonal anti-HRP Alexa Fluor 647 | Jackson ImmunoResearch | 123-605-021; RRID:AB_2338967 | 1/300 |

## Resource availability

### Lead contact
Further information and requests for resources should be directed to AN (nose@k.u-tokyo.ac.jp).

### Materials availability
This study did not generate any new reagents.

## Experimental model and subject details

### *D. melanogaster* strains
The following fly strains were used in this study.

- *yw* (Bloomington Drosophila Stock Center, #6598)
- *DWnt4^{C1}* (Null allele of *DWnt4* [*Cohen et al., 2002*], Bloomington Drosophila Stock Center, #6651)
- *DWnt4^{EMS23}* (Null allele of DWnt4 [*Cohen et al., 2002*], Kyoto Stock Center, #108-974)
- *DWnt5^{400}* (Null allele of DWnt5 [*Fradkin et al., 2004*], Bloomington *Drosophila* Stock Center, #64300)
- *DWnt6^{KO}* (Deletion of exon 1 of *DWnt6* [*Doumpas et al., 2013*], Bloomington Drosophila Stock Center, #76311)
- *DWnt8^{KO1}* (Knockout allele of *DWnt8* (a.k.a. *wntD*; [*Gordon et al., 2005*], Bloomington Drosophila Stock Center, #38407))
- UAS-*CD4-tdGFP* (A membrane-fused GFP, Bloomington Drosophila Stock Center, #35836)
- UAS-*CD4::GCaMP6f* (A membrane-fused GCaMP, *Takagi et al., 2017*)
- *R57C10-FlpL;;pJFRC201*-10XUAS-FRT>STOP > FRT-*myr::smGFP-HA, pJFRC240*-10XUASFRT >STOP > FRT-*myr::smGFP-V5-THS*-10XUAS-FRT>STOP > FRT-*myr::smGFP-FLAG, pJFRC210*-10XUAS-FRT>STOP>FRT-*myr::smGFP-OLLAS* (Named shortly as MCFO-6 in *Nern et al., 2015*, Bloomington Drosophila Stock Center [#64090], RRID:BDSC_64090)
- 20XUAS>dsFRT>-*CsChrimson::mVenus (attP18), pBPhsFlp2::Pest (attP3)*; Express CsChrimson::mVenus under GAL4 control when the STOP cassette is removed by hsFlp2, a heat-shock-dependent recombinase that targets FRT, *Takagi et al., 2017*

- *MB120B-spGAL4* (A combination of GAL4.AD and GAL4.DBD that specifically targets Wave neurons, *Takagi et al., 2017*)
- *R60G09-GAL4* (A GAL4 line that targets Wave neurons mainly in neuromeres A2-A6. Identified through visual screening of 6849 GAL4 driver lines registered in the FlyLight database [*Li et al., 2014*; *Manning et al., 2012*], Bloomington Drosophila Stock Center [#46441])
- *R77H11-GAL4* (A GAL4 line that targets Wave neurons mainly in neuromeres A1-A6. Described in *Masson et al., 2020*. Bloomington Drosophila Stock Center [#39983])
- *R77H11-LexA* (A LexA version of *R77H11* enhancer used for double labeling with *MB120B-spGAL4* and *R60G09-GAL4*. Bloomington Drosophila Stock Center [#54720])
- *Wnt4^{MI03717-Trojan-GAL4}* (A GAL4 insertion near the Wnt4 locus by MiMIC cassette recombination. Bloomington Drosophila Stock Center [#67449], RRID:BDSC_67449)
- *elav-GAL4^{3E1}* (Targets all neurons. Used to knock down *DFz2* in all neurons. *Davis et al., 1997*)
- UAS-*mCherry*.VALIUM10 (A control stock for RNAi reporter lines using VALIUM10 or VALIUM20 vectors. Generated by the Transgenic RNAi Project [TRiP], Bloomington Drosophila Stock Center, #35787, RRID:BDSC_35787)
- UAS-*DFz*-RNAi (An RNAi reporter line used to knock down DFz. Generated by the Transgenic RNAi Project (TRiP), Bloomington Drosophila Stock Center, #34321, RRID:BDSC_34321)
- UAS-*DFz2*-RNAi (An RNAi reporter line used to knock down DFz2. Generated by the Transgenic RNAi Project (TRiP), Bloomington Drosophila Stock Center, #67863, RRID:BDSC_67863)
- UAS-*DFz3*-RNAi (An RNAi reporter line used to knock down DFz3. Generated by the Transgenic RNAi Project (TRiP), Bloomington Drosophila Stock Center, #44468, RRID:BDSC_44468)
- UAS-*DFz4*-RNAi (An RNAi reporter line used to knock down DFz4. Generated by the Transgenic RNAi Project (TRiP), Bloomington Drosophila Stock Center, #64990, RRID:BDSC_64990)
- UAS-*drl*-RNAi (An RNAi reporter line used to knock down Drl. Generated by the Transgenic RNAi Project (TRiP), Bloomington Drosophila Stock Center, #39002, RRID:BDSC_39002)
- UAS-*Drl-2*-RNAi [TRiP] (An RNAi reporter line used to knock down Drl-2. Generated by the Transgenic RNAi Project (TRiP), Bloomington Drosophila Stock Center, #55893, RRID:BDSC_55893)
- UAS-*Drl-2*-RNAi [KK] (An RNAi reporter line used to knock down Drl-2. Generated by the VDRC genome-wide Drosophila RNAi project, Vienna Drosophila Resource Center, #102192)
- UAS-*smo*-RNAi (An RNAi reporter line used to knock down Smo. Generated by the Transgenic RNAi Project [TRiP], Bloomington Drosophila Stock Center, #43134, RRID:BDSC_43134)
- UAS-*Corin*-RNAi (An RNAi reporter line used to knock down Corin. Generated by the Transgenic RNAi Project [TRiP], Bloomington Drosophila Stock Center, #41721, RRID:BDSC_41721)
- UAS-*DFz2*-RNAi [GD] (An alternative RNAi reporter line used to knock down DFz2. Included in the GD collection [P-element] generated by the VDRC genome-wide Drosophila RNAi resource. VDRC #44390)
- UAS-*DFz2*-RNAi [KK] (An alternative RNAi reporter line used to knock down DFz2. Included in the KK collection [phiC31] generated by the VDRC genome-wide Drosophila RNAi resource. VDRC #108998)
- UAS-*DFz4*-RNAi [KK] (An alternative RNAi reporter line used to knock down DFz4. Included in the KK collection [phiC31] generated by the VDRC genome-wide Drosophila RNAi resource. VDRC #102339)
- UAS-*DFz2*-ORF (An ORF reporter line used for ectopic/overexpression of DFz2. Generated by the Zurich ORFeome Project, FlyORF, #F001187)
- UAS-*DFz4*-ORF (An ORF reporter line used for ectopic/overexpression of DFz4. Generated by the Zurich ORFeome Project, FlyORF, #F001662)
- LexAop-RCaMP2 (*Takagi et al., 2017*; DGRC #118796)

## Method details

### Immunohistochemistry

Immunohistochemistry experiments for 3rd instar larvae were performed as done previously (*Takagi et al., 2017*).

For 12–16 hr after egg laying (hAEL) embryos in *Figure 1—figure supplement 1B and C*, whole-mount staining was performed. First, the animals were chemically dechorionated with 50% sodium hypochlorite solution, transferred into a glass vial containing 2 mL heptane (organic layer), and were fixed by 2 mL addition of 4% paraformaldehyde in PBS (water layer) for 30 min at room temperature with shaking. After removal of the water layer, half of the heptane was removed, and 1 mL MeOH was added. The vial was vigorously shaken for 20 s to facilitate cracking of the vitelline membrane. After removing the organic layer, the embryos were washed with MeOH twice and with EtOH once. The

embryos were then transferred into a plastic microtube, and thereafter, the same protocol as in the larvae was applied.

For late-stage embryos in *Figure 1—figure supplement 1D*, *Figure 5*, and *Figure 5—figure supplement 2*, dissection was performed to increase staining efficacy. The dissection was performed by gluing the animal onto double-sided scotch tape and opening the dorsal part using a sharp needle.

The laser intensities used to quantify DWnt4 and DFz2 signals in *Figure 5—figure supplement 2* were kept the same between control and mutant embryos for each group.

The list of the antibodies and the dilution is as follows:

- rabbit anti-GFP (Af2020, Frontier Institute; 1:1000; RRID:AB_10615238)
- mouse anti-Fas2 (1D4, Hybridoma Bank [University of Iowa]; 1:10; RRID:AB_528235)
- rat anti-Elav (7E8A10, Hybridoma Bank [University of Iowa]; 1:10; RRID:AB_528218)
- mouse anti-Repo (8D12, Hybridoma Bank [University of Iowa]; 1:5; RRID:AB_528448)
- mouse anti-Brp (nc82, Hybridoma Bank [University of Iowa]; 1:100; RRID:AB_2392664)
- guinea pig anti-GFP (Af1180, Frontier Institute; 1:1000; RRID:AB_2571575)
- rabbit anti-HA (C29F4, Cell Signaling Technology; 1:1000; RRID:AB_1549585)
- mouse anti-V5 (R960-25, Invitrogen; 1:500; RRID:AB_2556564)
- rat anti-FLAG (NBP1-06712, Novus Biologicals; 1:500; RRID:1625981)
- rabbit anti-DWnt4 (*Cohen et al., 2002*; 1:100; kindly provided by M Sato)
- rabbit anti-DFz2 (*Packard et al., 2002*; 1:100; kindly provided by G Alegre, V Budnik, and T Thomson)
- goat Alexa Fluor 488 or Cy3-conjugated anti-rabbit IgG (A-11034 or A-10520, Invitrogen Molecular Probes; 1:300; RRID:AB_2576217 or AB_10563288)
- goat Alexa Fluor 555 or Cy5-conjugated anti-mouse IgG (A-21424 or A-10524, Invitrogen Molecular Probes; 1:300; RRID:AB_141780 or AB_2534033)
- goat Alexa Fluor 488-conjugated anti-rat IgG (A-11006, Invitrogen Molecular Probes; 1:300; RRID:AB_141373)
- goat Cy5-conjugated anti-HRP (123-175-021, 1:300, Jackson ImmunoResearch Laboratories Inc)
- Alexa Fluor 647 AffiniPure Goat Anti-HRP (123-605-021, 1:300, Jackson ImmunoResearch Laboratories Inc, RRID:AB_2338967)

Samples of poor quality (due to damage during dissection and/or drift during confocal imaging) were excluded from further analyses.

## Staging of embryos

Female and male flies of appropriate genotypes were kept in a vial for more than 1 day to allow copulation. The eggs were collected by trapping the parent flies on a yeast paste-coated apple plate, covered with a cup with small air holes. To prevent collecting withheld eggs (which are most likely pre-developed), the flies were allowed to lay their eggs on the plate for 1 hr. For experiments in *Figure 1—figure supplement 1*, we adopted the following procedure. The flies were transferred onto another fresh plate and were allowed to lay eggs for 1 hr. The plates were kept at 25°C until the embryos developed to the desired stage. The stages are indicated as hAEL, which includes an error of ±0.5 hr. The stages were further validated at the time of confocal imaging by observing the development of the gut (*Campos-Ortega and Hartenstein, 1985*), and animals that did not meet the standard were excluded.

For experiments in *Figure 5* and *Figure 5—figure supplement 2*, the flies were transferred onto a fresh plate and were allowed to lay eggs for 18–24 hr at 25°C. The stage (Stage 16) was confirmed by the development of the gut.

## Identification and characterization of *R60G09-GAL4* and embryonic Wave neurons

Since neurite outgrowth of larval neurons occurs during embryogenesis, we first searched for a GAL4 line that targets Wave neurons consistently from embryonic to larval stages, as the previously used GAL4 (*MB120B-spGAL4*) lacks expression at embryonic stages (animals with GFP-positive cells: *n*=2 out of 70 animals). We identified in the FlyLight database a sparse GAL4 line, *R60G09-GAL4*, which targets Wave neurons in 3rd instar larvae (*Figure 1—figure supplement 1E*) with the exception of a single pair of ascending neurons in T2 in the VNC, and also drives expression in two pairs of segmental neurons in the embryo (*Figure 1—figure supplement 1B and C*). Stage-by-stage observation revealed

that one of the GAL4-targeted embryonic neurons is a Wave neuron, which was continuously marked and thus can be traced through embryonic stages (10, 12, 16, 18, and 20 hAEL; *Figure 1—figure supplement 1B–D*) to larval stages (3rd instar; *Figure 1—figure supplement 1E*). Another *GAL4*-targeted non-Wave neurons (that reside posteriorly to Wave with the cell size being larger) started to get fragmented and densely stained in 16 hAEL (*Figure 1—figure supplement 1C*) and was extinct at later stages (*Figure 1—figure supplement 1D and E*) in segments anterior to A5, suggesting that this neuron is dMP2 (*Miguel-Aliaga and Thor, 2004*; *Miguel-Aliaga et al., 2008*). Although dMP2 neurons in segments A6-A8 are known to survive up to larval stages, the GAL4-driven expression in dMP2 neurons vanished in these stages (*Figure 1—figure supplement 1E*).

## Primary RNAi/mutant tests

To visualize the Wave neurites, membrane-bound reporters (CD4-tdGFP or CD4-GCaMP6f, with the latter being brighter when stained) were expressed. The CNS was stained by anti-GFP and anti-Fas2 antibodies. Confocal stack images of the CNS were taken by using a confocal microscope (FV1000, Olympus).

For RNAi-based KD experiments (*Figure 1—figure supplement 2D*), female UAS-RNAi flies were crossed to *R60G09-GAL4. UAS-CD4-GCaMP6f* males and the larvae of the next generation were examined. The nine genes tested are known guidance cue receptors possibly involved in Wnt signaling: *DFz, DFz2, DFz3, DFz4, drl, Drl-2, smo,* and *Corin*. All the RNAi lines used here are from the Transgenic RNAi Project (TRiP), except for *Drl-2*, where TRiP was used for one sample and KK line for the other three.

For mutant-based knockout experiments (*Figure 1—figure supplement 2E*), R77H11-GAL4, UAS-CD4-GCaMP6f (or MB120B-spGAL4, UAS-CD4-GCaMP6f for DWnt5 mutants), flies were used since the expression was nicely confined to Wave neurons (*Masson et al., 2020*).

## Single-Wave labeling by heat-shock FlpOut (*Figure 4D–G*)

Female *20XUAS>dsFRT>-CsChrimson::mVenus, pBPhsFlp2::Pest; DWnt4^{EMS23}* flies were crossed to *DWnt4^{C1}; R77H11-GAL4* males and housed in tubes with fly food. The tubes containing eggs and 1st to 2nd instar larvae were placed into an incubator set to 37.5–38.7°C for 1 hr. The tubes were put back to 25°C to raise the eggs up to the 3rd instar larvae. The CNS was dissected and Wave neurons were visualized by staining mVenus with anti-GFP antibody. The segment identities of Wave neurons were determined by observing the entry point of the neurite from the soma, based on anti-HRP staining.

## Optogenetics in freely moving larvae (*Figure 6*)

*UAS-CsChrimson* female flies were crossed to *R60G09-GAL4* or *UAS-DFz2-RNAi [TRiP]; R60G09-GAL4* males. The larvae from the next generation were grown at 25°C. 2nd or 3rd instar larvae were picked, gently washed, and transferred onto an apple juice agar plate coated with yeast paste, either containing 2 mM of all-trans retinal (ATR) or none (ATR concentration was calculated based on the volume of the dry yeast. Note that the same amount of distilled water was added to make yeast paste). The plate was covered with plastic cover and aluminum foil and placed at 25°C for one night. The behavioral experiment was conducted on an apple juice agar plate, which was placed on a heating plate to set the surface temperature of the agar within 25°C ± 1°C. The larvae were placed on the fresh apple plate for over 5 min before the behavioral assays. 660 nm LED light at the density of 20–25 µW/mm$^2$ (Thorlabs) was used for the stimulation of CsChrimson. The stimulation trials were delivered twice for each animal, with a duration of 10–15 s for each trial and >15 s intervals between each trial. Video recording was conducted under a stereomicroscope (SZX16, Olympus), while the background illumination was minimized so as not to activate CsChrimson.

## Gentle touch assay (*Figure 7*)

The behavioral experiment was conducted on an apple juice agar plate, which was placed on a heating plate to set the surface temperature of the agar within 25°C ± 1°C. von Frey filament (Touch Test Sensory Evaluator [0.07 g], North Coast; #NC12775-04) was used to provide gentle head touch. A gentle touch was applied to the target site (dorsal side of T3 ±1 segment) until the filament showed a mild bend, so that a consistent force was applied across trials. Trials with failed stimulation (wrong

location, insufficient filament bend) were excluded from the analyses. Five trials were performed for eight animals in each group. The experiment was performed by mixing the identities of genetic groups to make their identities blind to the examiner. The genotypes are as follows: driver control (*yw* × *R60*G09-GAL4), TRiP effector control (*yw* × *UAS-DFz2-RNAi*[TRiP]), KK effector control (*yw* × UAS-*DFz2-RNAi*[KK]), GD effector control (*yw* × UAS-*DFz2-RNAi*[GD]), TRiP experimental (UAS-*DFz2-RNAi*[TRiP] × *R60*G09-GAL4), KK experimental (UAS-*DFz2-RNAi*[KK] × *R60*G09-GAL4), GD experimental (UAS-*DFz2-RNAi*[TRiP] × *R60*G09-GAL4).

## Quantification and statistical analysis

### Primary RNAi/mutant test (*Figure 1—figure supplement 2*)

The projection of Wave axons was manually examined with Fas2 reference as follows. For an axon, as it runs dorsally, one of the Fas2 tract TP1 was considered as a boundary (*Landgraf et al., 2003b*). The neuromere identity was determined starting from T3, where Fas2 tracts are uniquely identifiable. For analysis in *Figure 1—figure supplement 2G, G'*, HRP staining was used as a reference and hence the criteria used was the same as in the clonal analysis (described below).

For examination of the Wave axon/dendrite projection region, three animals were examined for each RNAi strain. If GFP expression was not obtained for more than three neurons (due to GAL4 expression stochasticity), we added three to six animals until the sample size reached three.

### Clonal analyses (*Figures 1–4* and *Figure 1—figure supplements 2 and 3*)

Wave neurites were visualized by anti-V5/anti-HA/anti-FLAG (for MCFO) or by anti-GFP (for heat-shock FlpOut) antibody staining with the anti-HRP counterstain, which clearly stains the anterior commissure (AC) and posterior commissure (PC) in each neuromere. The segment identity was assigned starting from T3, which is uniquely identifiable. The boundary between neuromeres $A_k$ and $Ak_{k+1}$ ($k$=1, 2, …, 7) is approximately the centerline between the PC in neuromere $A_k$ and AC in neuromere $Ak_{k+1}$ (*Landgraf et al., 2003a*). The presence of a neurite in a specific neuromere was determined as follows. For anterior projection, when the neurite crossed the PC in neuromere $A_k$, it was counted as being present in $A_k$. For posterior projection, when the neurite crossed the AC in neuromere $A_k$, it was counted as being present in $A_k$. These definitions were intended to exclude vague projections into a neuromere. Quantification of relative axon length is described in the following section.

### Quantification of relative axon length (*Figures 1, 3, and 4* and *Figure 1—figure supplement 2*)

The relative position $c$ ($0 \leq c < 1$) of the axon landmarks (axon branching point, anterior end, and posterior end) was calculated as follows.

$$c\left(x, y\right) = \frac{\left(x_1 - x_0\right)\left(x - x_0\right) + \left(y_1 - y_0\right)\left(y - y_0\right)}{\left(x_1 - x_0\right)^2 + \left(y_1 - y_0\right)^2}$$

where point ($x$, $y$) denotes the pixel coordinate of a specific point in a confocal stack image taken from the dorsal side. The points of interest are:

($x$, $y$): axon branching point, anterior end, or posterior end

($x_0$, $y_0$): the anterior segmental border (calculated as the midpoint of the center of the dorsal posterior commissure [dPC] of the prior neuromere and the center of the dorsal anterior commissure [dAC] of the present neuromere, derived from anti-HRP staining)

($x_1$, $y_1$): the posterior segmental border (calculated as the midpoint of the center of the dPC of the present neuromere and the center of the dAC of the next neuromere, derived from anti-HRP staining).

By adding relative position $c$ to the neuromere identity $n$ ($n = 1, 2, \ldots, 11$ for neuromeres T1, T2, T3, A1, …, A8), normalized landmark positions $\hat{c}(x, y) = c(x, y) + n(x, y)$ were defined and their distances were used to calculate relative axon lengths.

As relative positions of landmarks are critical for these analyses, images with drifts and/or perturbation were excluded.

## Quantification of DWnt4/DFz2 expression (*Figure 5* and *Figure 5—figure supplements 1 and 2*)

To quantify the fluorescence from immunohistochemistry images, ROIs were set to each hemisegment to calculate the mean intensity. The segment identity was assigned for each block of anterior and posterior commissures (AC and PC) in the neuropil starting from A8 (in embryos) or A1 (in larvae).

To quantify the anti-DWnt4 and anti-DFz2 signals, Z-stack images that extracted the maximum intensity pixels within the neuropil in each hemisegment were used. ROIs were set as circles that cover the lateral neurite-dense regions in left and right hemisegments. The signals from anti-DWnt4/DFz2 were normalized by the anti-HRP signals in the corresponding ROIs. Neuromere A1 was excluded from the analysis as many images did not cover this region. Samples with ambiguous A-P directionality were also excluded.

To quantify the GFP.nls and anti-Elav signals in embryos, the ROIs were set as polygons (drawn based on the anti-Elav image) that encircled the segmental bumps at the lateral edge and imaginary boundaries between segments (that appeared as arc-shaped dark ditches). Since the boundary was less evident posterior than A8, all the terminal segments were included and named as 'A8/9'.

## Quantification of larval behaviors (*Figures 6 and 7*)

Larval behaviors were manually quantified. Larval locomotion was counted when a complete sequence of muscle contraction across all segments was observed, with the direction from anterior to posterior being backward locomotion and the other way around being forward locomotion (*Berni et al., 2012*). Rolling was defined as a 360-degree rolling of larval body to the lateral direction.

In the quantification of larval behaviors upon optogenetic stimulation (*Figure 6*), the first 10 s just prior to and after stimulation onset in each trial were used (to avoid the effect of desensitization) and then summed across trials.

In the quantification of stride durations (*Figures 6 and 7*), the duration between the start (landing of the posterior end of the larval body) and the end (landing of the anterior end of the larval body) of forward locomotion that occurred just prior to and after the first stimulation onset (*Figure 6*) or just after each gentle touch (*Figure 7*) was measured. In *Figure 7*, the trials where the larvae were outside the field of view when performing forward locomotion following the stimulus were excluded from the analyses.

In the quantification of behaviors upon gentle touch (*Figure 7*), the Kernan scoring was performed by manually observing the recorded video. To minimize scoring bias, two independent analyses were performed (by SM and S Takagi), and we confirmed that a consistent conclusion was obtained.

## Statistical analyses

All statistical tests were performed using R (https://www.r-project.org/). The statistical tests used are indicated in the respective figure legends or in the text.

In the primary RNAi analysis and the mutant analysis, the Chi-square test followed by Haberman's adjusted residual analysis (*Haberman, 1973*) was applied to determine the statistical significance of each cell in the cross-classified table. To avoid multiple comparison problems, adjusted significance level was calculated in accord with *Sidak, 1967*, as is recommended in *Beasley and Schumacker, 1995*, as follows:

$$\alpha_{adjusted} = 1 - \left(1 - \alpha\right)^{\frac{1}{k}}$$

where $k$ denotes the number of cells in the table. Here, we set $\alpha = 0.05$.

## Acknowledgements

We would like to thank Shoya Ohura and Sawako Niki for the help in screening, Hiroshi Kohsaka for sharing the code for behavior analysis, James Truman for providing reagent identifier, and Matthias Landgraf for sharing unpublished observations. We thank Richard Benton and Michael P Shahandeh for comments on the earlier version of the manuscript. We would also like to thank Gimena Alegre, Vivian Budnik, Andrea Brand, Tzumin Lee, Makoto Sato, Tetsuya Tabata, Travis Thomson, Bloomington Drosophila Stock Center, Vienna Drosophila Resource Center, Developmental Studies Hybridoma

Bank, FlyORF team, and KYORIN-Fly for providing reagents. This work was supported by MEXT/JSPS KAKENHI grants (15H04255, 18H05113, 19H04742, 20H05048, 21H02576, 21H05675, 22H05487, 22K19479, 23H04213, 24H01225, 24K02117, 25H02486 to AN; 18J10483 to S Takagi).

## Additional information

### Funding

| Funder | Grant reference number | Author |
|---|---|---|
| MEXT/JSPS KAKENHI | 15H04255 | Akinao Nose |
| MEXT/JSPS KAKENHI | 18J10483 | Suguru Takagi |
| MEXT/JSPS KAKENHI | 24K02117 | Akinao Nose |
| MEXT/JSPS KAKENHI | 25H02486 | Akinao Nose |
| MEXT/JSPS KAKENHI | 24H01225 | Akinao Nose |
| MEXT/JSPS KAKENHI | 23H04213 | Akinao Nose |
| MEXT/JSPS KAKENHI | 22K19479 | Akinao Nose |
| MEXT/JSPS KAKENHI | 22H05487 | Akinao Nose |
| MEXT/JSPS KAKENHI | 21H05675 | Akinao Nose |
| MEXT/JSPS KAKENHI | 21H02576 | Akinao Nose |
| MEXT/JSPS KAKENHI | 20H05048 | Akinao Nose |
| MEXT/JSPS KAKENHI | 19H04742 | Akinao Nose |
| MEXT/JSPS KAKENHI | 18H05113 | Akinao Nose |

The funders had no role in study design, data collection and interpretation, or the decision to submit the work for publication.

### Author contributions

Suguru Takagi, Conceptualization, Data curation, Formal analysis, Investigation, Visualization, Writing – original draft, Writing – review and editing, Data acquisition: Figures 1, 3, 4A–C, 5A, B, 6, Figure 1–figure supplement 1, 2A–E, 3, Figure 5–figure supplement 1, Data analysis, plotting, and depositing; Shiina Takano, Investigation, Data acquisition: Figure 1–figure supplement 2F and F'; Tomohiro Kubo, Investigation, Data acquisition: Figure 2; Yusaku Hashimoto, Investigation, Data acquisition: Figure 1–figure supplement 2G and G'; Shu Morise, Investigation, Data acquisition: Figures 4D–G and 6; Xiangsunze Zeng, Investigation, Data acquisition: Figure 1–figure supplement 2F and F'; Akinao Nose, Conceptualization, Supervision, Funding acquisition, Writing – original draft, Project administration, Writing – review and editing, Data acquisition: Figures 5C–F and Figure 5–figure supplement 2

### Author ORCIDs

Suguru Takagi ⓘ https://orcid.org/0000-0002-6381-7152
Akinao Nose ⓘ https://orcid.org/0000-0002-0526-2128

Reviewer #1 (Public review): https://doi.org/10.7554/eLife.98624.3.3.sa1
Reviewer #2 (Public review): https://doi.org/10.7554/eLife.98624.3.3.sa2
Author response https://doi.org/10.7554/eLife.98624.3.3.sa3

## Additional files

### Supplementary files
MDAR checklist

## Data availability

Raw data and analysis codes that are used for the present study are available at Mendeley Data.

The following dataset was generated:

| Author(s) | Year | Dataset title | Dataset URL | Database and Identifier |
|---|---|---|---|---|
| Suguru T | 2025 | Segment-specific axon guidance by Wnt/Fz signaling diversifies motor commands in Drosophila larvae (Takagi et al.) | https://doi.org/10.17632/66jh2hxz2w | Mendeley Data, 10.17632/66jh2hxz2w |

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
