## [Editor Report · eLife Assessment]

The study by Takagi and colleagues is an **important** contribution to the question of how homologous neuronal circuits might be wired differently to elicit specific behaviours. The authors combine genetic, neuroanatomical, and behavioral data to provide **convincing** evidence that Dfz2/DWnt4 signaling controls the innervation pattern of wave command neurons in the fly larva, and thereby behavioral locomotion program selection.

---

## [Referee Report · Reviewer #1 (Public review)]

Summary

In this study Takagi and colleagues demonstrate that changes in axonal arborization of the segmental wave motor command neurons are sufficient to change behavioral motor output.

The authors identify the Wnt receptors DFz2 and DFz4 and the ligand Wnt4 as modulators of the stereotypic segmental arborization pattern of segmental wave neurons along the anterior-posterior body axis. Based on both embryonic expression pattern analysis and genetic manipulation of the signaling components in wave neurons (receptors) and the neuropil (Wnt4) the authors convincingly demonstrate that Wnt4 acts as a repulsive ligand for DFz2 that restricts posterior axon guidance of both anterior and posterior wave neurons. They also provide first evidence that Wnt4 potentially acts as an attractive ligand for Df4 to promote posterior extension of p-wave neurons. Interestingly, artificial optogenetic activation of all wave neurons that normally induces a backward locomotion due to the activity of anterior wave neurons, fails to induce backward locomotion in a DFz2 knock down condition with altered axonal extensions of all wave neurons towards posterior segments. In addition, the authors now observe enhanced fast forward locomotion a feature normally induced by posterior wave neurons. Consistent with these findings, they observe that the natural response to an anterior tactile stimulus is similarly altered in DFz2 knock down animals. The animals respond with less backward movement and increase fast forward motion. These results suggest that alterations in the innervation pattern of wave motor command neurons are sufficient to switch behavioral response programs.

Strengths

The authors convincingly demonstrate the importance of Wnt signaling for anterior-posterior axon guidance of a single class of motor command neurons in the larval CNS. The demonstration that alteration of the expression level of a single axon guidance receptor is sufficient to not only alter the innervation pattern but to significantly modify the behavioral response program of the animal provides a potential entry point to understand behavioral adaptations during evolution.

Weaknesses

The authors demonstrate an alteration of the behavioral response to a natural tactile stimulus and correlate this to morphological alterations observed in the single-neuron analyses. As the authors suggest an alteration of the command circuitry, a direct observation of the downstream activation pattern in response to selective optogenetic stimulation of anterior wave neurons (if possible with appropriate genetic tools in the future) would further strengthen their claims.

---

## [Referee Report · Reviewer #2 (Public review)]

Summary:

In the manuscript, the authors aim to determine the molecular mechanisms involved in wiring the segmentally homologous a- and p -Wave neurons distinctively and thus are functionally different in modulating forward or backward locomotion. The genetic screen focused on Wnt/Fz-signaling due to its known anterior-to-posterior guidance roles in mammals and nematodes.

Strengths:

The conclusion that Frizzled receptors DFz2 and DFz4 as well as the DWnt4 ligand is essential for normal segment-specific axon projections of Wave command neurons is strongly supported by the elaborate morphological analyses of numerous Wnt/Fz in gain and loss of function mutants. The distinctive Wnt/Fz ligand-receptor gradients also imply that they contribute to the diversification of Wave neurons in a location-dependent manner and that DFz2 and DFz4 may have opposing effects on axon extension.

Labeling of synaptic marker Bruchpilot in DFz2 mutants in this revised manuscript, now supports that the ectopic projections in a-Wave neurons make synaptic connections. Finally, the altered responses in two behavioral assays (optogenetic stimulation of all Wave neurons or tactile stimuli on heads using a von Frey filament) further strongly support the main conclusion, that Wnt/Fz-signaling is essential for the guidance of both Wave neurons and in diversifying their protection pattern in a segment-specific manner.

Weaknesses:

There are no major weaknesses in the revised version of this work.

Re-analysis of DFz2 expression now shows it is bidirectionally distributed. This new result does not affect the previous and current conclusions for the a-Wave neurons but leaves alternative interpretations for p-Wave neurons, which the author now included in their discussions. Evidently, it seems unlikely that the complex wiring of the numerous segmental a- and p-Wave neurons will be solely dependent on Wnt4-DFz2/4 but are likely to also involve other Wnt/Fz (see, Figure 1-figure supplement 2) or distinct guidance signaling pathways. However, unraveling all factors involved is certainly beyond the scope of this study, and the main conclusions made by the authors are well supported by the data provided.

---

## [Author Response]

The following is the authors’ response to the original reviews.

**Reviewer #1 (Public Review):**
SummaryIn this study, Takagi and colleagues demonstrate that changes in axonal arborization of the segmental wave motor command neurons are sufficient to change behavioral motor output.The authors identify the Wnt receptors DFz2 and DFz4 and the ligand Wnt4 as modulators of stereotypic segmental arborization patterns of segmental wave neurons along the anterior-posterior body axis. Based on both embryonic expression pattern analysis and genetic manipulation of the signaling components in wave neurons (receptors) and the neuropil (Wnt4) the authors convincingly demonstrate that Wnt4 acts as a repulsive ligand for DFz2 that restricts posterior axon guidance of both anterior and posterior wave neurons. They also provide the first evidence that Wnt4 potentially acts as an attractive ligand for Df4 to promote the posterior extension of p-wave neurons. Interestingly, artificial optogenetic activation of all wave neurons that normally induces backward locomotion due to the activity of anterior wave neurons, fails to induce backward locomotion in a DFz2 knockdown condition with altered axonal extensions of all wave neurons towards posterior segments. In addition, the authors now observe enhanced fast-forward locomotion, a feature normally induced by posterior wave neurons. Consistent with these findings, they observe that the natural response to an anterior tactile stimulus is similarly altered in DFz2 knockdown animals. The animals respond with less backward movement and increased fast forward motion. These results suggest that alterations in the innervation pattern of wave motor command neurons are sufficient to switch behavioral response programs.StrengthsThe authors convincingly demonstrate the importance of Wnt signaling for anteriorposterior axon guidance of a single class of motor command neurons in the larval CNS. The demonstration that alteration of the expression level of a single axon guidance receptor is sufficient to not only alter the innervation pattern but to significantly modify the behavioral response program of the animal provides a potential entry point to understanding behavioral adaptations during evolution.WeaknessesWhile the authors demonstrate an alteration of the behavioral response to a natural tactile stimulus the observed effects, a reduction of backward motion and increased fast-foward locomotion, currently cannot be directly correlated to the morphological alterations observed in the single-neuron analyses. The authors do not report any loss of innervation in the "normal" target region but only a small additional innervation of more posterior regions. An analysis of synaptic connectivity and/or a more detailed morphological analysis that is supported by a larger number of analyzed neurons both in control and experimental animals would further strengthen the confidence of the study. As the authors suggest an alteration of the command circuitry, a direct observation of the downstream activation pattern in response to selective optogenetic stimulation of anterior wave neurons would further strengthen their claims (analogous to Takagi et al., 2017, Figure 4).

We sincerely thank the reviewer for their insightful comments, which were instrumental in improving our manuscript. In response to the reviewers’ suggestion, we have now studied Brp expression and demonstrate that the ectopically extending Wave axons in the posterior region do contain synapses (new Figure 2). This finding supports the idea that these axons are functionally connected to ectopic downstream circuits.

Additionally, we have increased the number of analyzed Wave clones in Figure 1F-J (WT and DFz2 KD) and new Figure 3C-G (WT; formerly Figure 2C-G) to strengthen the morphological analyses. We fully agree with the reviewer that “direct observation of the downstream activation pattern in response to selective optogenetic stimulation” would further reinforce our conclusions. However, this was not feasible in the current study since we found that the Wave-Gal4 driver used in this study, which drives expression during embryonic stages, does not drive sufficiently strong expression in the larvae to enable selective optogenetic stimulation (please see below for details).

**Reviewer #2 (Public Review):**
Summary:The authors previously demonstrated that anterior-located a-Wave neurons (neuromeres A1-A3) extend axons anteriorly to connect to circuits inducing backward locomotion, while p-Wave axon (neuromeres A4-A7) project posteriorly to promote forward locomotion in Drosophila larvae. In the manuscript, the authors aim to determine the molecular mechanisms involved in wiring the segmentally homologous Wave neurons distinctively and thus are functionally different in modulating forward or backward locomotion. The genetic screen focused on Wnt/Fz-signaling due to its known anterior-to-posterior guidance roles in mammals and nematodes.Strengths:Knock-down (KD) DFz2 with two independent RNAi-lines caused ectopic posterior axon and dendrite extension for all a- and p-Wave neurons, with a-Wave axon extending into regions where p-Wave axons normally project. Both behavioral assays (optogenetic stimulation of all Wave neurons or tactile stimuli on heads using a von Frey filament) show that backward movement is reduced or absent and that the speed of evoked fast-forward locomotion is increased. This demonstrates that altered projections of Wave do alter behavior and the DFz2 KD phenotype is consistent with the potential aberrant wiring of a-Wave neurons to forward locomotion-promoting circuits instead of to backward locomotion-promoting circuits.The main conclusion, that Wnt/Fz-signaling is essential for the guidance of Wave neurons and in diversifying their protection pattern in a segment-specific manner, is further supported by the results showing that DFz2 gain of function causes shortening of a-Wave but not p-Wave axon extensions towards the posterior end and that KD of DFz4 causes axonal shortening only in A6-p-Wave neurons but does not affect dendrites or processes of other Wave neurons. A role for ligand Wnt4 is demonstrated by results indicating that WNT4 mutants' posterior extension of aWave axons was elongated similar to DFz2 KD animals and p-Wave axon extension towards the posterior end was shortened similar to DFz2 KD animals. Finally, a DWnt4 gradient decreasing from the posterior (A8) to the anterior end (A2), similar to that described in other species, is supported by analyses of DWnt4 gene expression (using Wnt4 Trojan-Gal4) and protein expression (using antibodies). In contrast, DFz2 receptor levels seemed to decrease from the anterior (A2) to the posterior end (A5/6). Together the results support the conclusion that opposing Wnt/Fz ligand-receptor gradients contribute to the diversification of Wave neurons in a location-dependent manner and that DFz2 and DFz4 have opposing effects on axon extension.Weaknesses:Wave axon and dendrite projections are not exclusively determined by Wnt4, DFz2, and DFz4, and are likely to involve other Fz receptors, Wt ligands, and other types of receptor-ligand signaling pathways. This is in part supported by the fact that Wnt4 loss of function also resulted in phenotypes that do not mimic DFz2 KD or DFz4 KD (Figures 3D, E, and F) and that other Fz/Wnt mutants caused wave neuron phenotypes (Figure 1-supplement 2, D+E). This is not a weakness per se, since it doesn't affect the main conclusion of the manuscript. However, the description and analyses of the data in particular for Figure 1-supplement 2 D should be clarified in the legend. The number within the bars and the asterisks are not defined. It's presumed they refer to numbers of animals assessed and the asterisk next to DFz2 and DFz4 indicate statistically significant differences. However, only one p-value is provided in the legend. It is also unclear if p-values for the other mutants have not been determined or are non-significant. At least for mutants like Corin, which also exhibit altered axon projections, the p-values should be provided.

We appreciate this reviewer’s careful attention to detail and intellectual curiosity. We apologize for the confusions caused by the statistical reporting in Figure 1 – figure supplement 2D. The numbers shown in the bars represent the number of neurons (i.e. Wave neurons from left or right hemisphere). As mentioned in Materials and Methods section, we applied Chi-square test followed by Haberman's adjusted residual analysis to determine the statistical significance of each RNAi group. The p-value provided in the figure legend corresponds to the Chi-square test. P-values for Haberman's adjusted residual analysis were calculated for all RNAi groups and groups without the asterisk are not statistically significant. We have clarified these points in the corresponding figure legend.

Figure 4 D, F. The gradient for Wnt4 was determined by comparison of expression levels of other segments to A8 but the gradient for DFz2 was by comparison to A2 and the data supports opposing gradients. However, for DFz2 (Figure 4, F) it seems that the gradient is bi-directional with the lowest being in A5 and increasing towards A2 as well as A8. Analysis should be performed in reference to A8 as well to determine if it is indeed bi-directional. While such a finding would not affect the interpretation of aWave neurons, it may impact conclusions about p-Wave neuron projections.

We thank the reviewer for highlighting this interesting possibility. In response, we performed an additional analysis of the DFz2 gradient by comparing the signal from each neuromere to that from A8 (new Figure 5—figure supplement 3). This analysis confirmed that the gradient is indeed bidirectional. We revised the description of DFz2 expression accordingly in the revision. We believe this finding does not affect our main conclusions since only the anterior gradient is relevant for a-Wave axon guidance.

As discussed above, the DFz2 KD phenotypes are consistent with the potential aberrant wiring of a-Wave neurons to forward locomotion-promoting circuits instead of to backward locomotion-promoting circuits. However, since the axon and dendrites of a-Wave and p-Wave are affected the actual dendritic and axonal contributions for the altered behavior remain elusive. The authors certainly considered a potential contribution of altered dendrite projection of a-Wave neurons to the phenotype and their conclusion that altered axonal projections are involved is supported by the optogenetic experiment "bypassing" sensory input (albeit it seems unlikely that all Wave neurons are activated simultaneously when perceiving natural stimuli).However, the author should also consider that altered perception and projection of pWave neuron may directly (e.g. extended P-wave axon projections increase forward locomotion input thereby overriding backward locomotion) or indirectly (e.g. feedback loops between forward and backward circuits) contribute to the altered behavioral phenotypes in both assays. It is probably noteworthy that the more complex behavioral alterations observed with mechanical stimulation are likely to also be caused by altered dendritic projections.

We fully agree with the reviewer’s thoughtful interpretation. We have now included these important possibilities in the revised Discussion section. Specifically, we acknowledge that while the DFz2 knockdown phenotypes are consistent with aberrant wiring of a-Wave neurons to forward locomotion-promoting circuits, the contributions of both axonal and dendritic alterations remain unclear. We also recognize that altered perception and projection of p-Wave neurons may directly or indirectly contribute to the observed behavioral phenotypes, particularly in response to mechanical stimulation.

Presynaptic varicosities of a-Wave neurons in DFz2 KD animals are indicated by orange arrows in Figure 1. However, no presynaptic markers have been used to confirm actual ectopic synaptic connections. At least the authors should more clearly define what parameters they used to "visually" define potential presynaptic varicosities. Some arrows seem to point to more "globular structures" but for several others, it's unclear what they are pointing at.

As mentioned in our response to Reviewer #1, we have now performed Brp immunostaining to confirm the presence of ectopic synaptic connections (new Figure 2). This analysis supports the interpretation that the presynaptic varicosities observed in DFz2 knockdown animals represent actual synaptic sites. We also clarified in the figure legend the visual criteria used to identify potential presynaptic varicosities.

**Reviewing Editor (Recommendations For The Authors):**
There are a few major concerns that we recommend the authors address:(1) Neuroanatomy: The point aberrant synaptic connectivity of a-Wave neurons following Dfz2 knockdown could be substantiated. This could be done by using a presynaptic marker and showing ectopic posterior presynaptic sites (and/or reduced anterior presynaptic sites) in a-wave neurons.

As mentioned in our response to the public review, we now have used Brp as a presynaptic marker to quantify the number and distribution of presynaptic sites along the normal and ectopic a-Wave axons (new Figure 2). We show that ectopic posterior Wave axons do contain presynaptic sites.

(2) Gradient calculations: As detailed in the reviews below, the Dfz2 gradient looks like it may be bidirectional. Changing the way the gradient is calculated might help address this point.

As mentioned in our response above, we now have recalculated the gradient by comparing the DFz2 signal to A8 and show that it indeed is bidirectional (new Figure 5—figure supplement 2; formerly Figure 4—figure supplement 2).

(3) Statistics and sample sizes: As detailed in the reviews, some of the statistical reporting could be improved. Further, increasing sample sizes could help bolster confidence in the data as well.

As mentioned above, we have added a description on the sample size, asterisks, and p-values in Figure 1 – figure supplement 2 legend. We also increased sample sizes of single Wave neurons in control and DFz2 knock-down animals (Figure 1F-J (WT and DFz2 KD) and new Figure 3C-G (WT; formerly Figure 2C-G)).

(4) It would help to include some discussion of the potential contributions of altered p-wave neurons to the observed phenotypes.

As described above, we have added in the Discussion potential contributions of altered p-wave neurons to the observed phenotypes.

**Reviewer #1 (Recommendations For The Authors):**
(1) In the current model the authors assume that posterior elongation of a-wave neuron connectivity (axonal projections) induces a loss of connectivity to their natural targets, as backward motion is no longer induced, and a gain of connectivity to posterior wave neuron targets. Is this at the cost of innervation of p-wave neurons, e.g. did these neurons now lose connectivity to their natural targets as well? Therefore, it would be very interesting if the authors would test the behavioral responses to tactile stimuli in the posterior parts of the animal - does the response pattern change?

This is indeed an interesting possibility that p-Wave function is altered upon DFz2 knock-down and hence behavioral response to posterior touch is changed. However, it is technically challenging to test this with tactile stimuli, due to the difficulty of (1) distinguishing between normal and fast-forward locomotion and (2) delivering a posterior touch stimulus while the larva is moving forward, which is the default behavior of the larvae on an agar plate.

As highlighted above, the authors should provide additional evidence that the circuit response to a-wave neurons is changed after a DFz2 knockdown. The authors should monitor the activation wave in response to optogenetic activation of anterior wave neurons - analogous to the data provided in Figure 4 of their 2017 paper. If this response is now switched for a-wave activation but not p-wave activation it would greatly support their claims and this data would be less ambiguous compared to the behavioral locomotion data.

As described in our response to the public review, we attempted this approach but found that the in vitro optogenetics experiment is unfortunately not feasible due to relatively weak expression of R60G09-GAL4 in the larvae. Local activation of control aWave induced fictive backward locomotion only at low frequencies, making comparison with the experimental a-Wave very difficult. The MB120B-spGAL4 used in our 2017 study could not be employed in this study as it does not drive expression during the embryonic stages and thus cannot be used to knock down DFz2 during development.

(2) Related to this point. Why would the normal "backward" circuitry of a-wave neurons be functionally suppressed in Dfz2 knockdowns? Do the authors observe reduced synaptic connectivity in these segments? Vesicle clustering of synaptotagmin or other presynaptic markers could be used as a first. As the innervation pattern is only extended by approximately one segment, it is surprising that the changes are so significant.

We agree that these are important and interesting points, which remain to be explored in the future study. As described above, we have performed Brp immunostaining and showed that the posterior ectopic axons of a-Wave do contain synapses (new Figure 2). We also found a slight decrease in the number of synapses in the anterior region, which could partially contribute to the weaker activation of downstream neurons responsible for eliciting backward locomotion. Another possibility is that backward suppression occurs through lateral interaction among downstream circuits. Since forward and backward locomotion do not occur simultaneously, it is likely that the circuits driving these two behaviors are mutually inhibitory. Upon DFz2 knock down in a-Wave, downstream neurons inducing fastforward locomotion may become more strongly activated than those inducing backward locomotion, resulting in inhibition of the latter via a “winner-take-all” mechanism. Since these discussions are highly speculative, we chose not to include them in the revised manuscript.

(3) The low number of neurons analyzed per segment is of slight concern. This is particularly the case for the control data set used in Figure 1 and Figure 2. As stated, the same datasets are used for both figures. However, at most 6 neurons were analyzed (and for two segments only 3). The control morphology may be more variable than indicated by this data.

As mentioned above, we now have dissected 50 larvae each for the control and experimental groups, obtained seven and six clones respectively, and included these data in the revised manuscript. We apologize that the sample sizes are still relatively small but hope the reviewer understands the inherently low “hit rate” of the stochastic labelling method.

It is somewhat curious that in Figure 1- Supplement 3 the authors report the same number of control clones per segment as in Figure 1/2 - is this simply a coincidence? And if this is an independent dataset why did the author use new controls here but not for Figure 2? It is clear that it is very difficult to generate this data but increasing the n-number beyond 3-6 per segment would significantly increase the confidence in the presented data.

We apologize for the confusion. The data in Figure 1 – figure supplement 3 represent the innervation pattern of dendrites, not axons. We have corrected the figure caption accordingly. These data were obtained from the same samples used to analyze axonal innervation, as shown in the original version of Figure 1F-J.

(2) The name of the RNAi lines should be indicated in Figure 1 and Figure Supplement 3 to facilitate reading - at least the precise names should be given in both figure legends.

We have added these labels in the revised figure legends as requested.

(3) In Figure 4E again the control numbers of Figure 1 for the A2-wave axon are reused. This does not seem appropriate as now a different Gal4 driver is used and a different method to induce individual neuronal clones. Both components may induce significant variability in expression or arborization. As only 3 clones for the wnt4 mutant condition are analyzed (and compared to 5 control clones), this data does not allow for strong conclusions. The authors clearly state the reuse and different methods in the legend of Figure 4 F/G but should also highlight it for the E panel.

Here, we assume that the reviewer is referring to the former Figure 3 (now Figure 4). We have added a note in the legend that the control data, obtained using a different method, were reused in this panel.

(4) The expression levels of DWnt4 and DFz2 were analyzed at the end of embryogenesis. At what developmental stage does the axonal extension of wave neurons take place? Is the gradient maintained throughout the first larval stages?

Based upon the lateral view of Wave neurons in Figure 1—figure supplement 1D, we think that the axonal extension is already established by approximately 20 hr after egg laying. Previously, we performed Wnt4^MI03717-Trojan-GAL4^ > GFP.nls immunostaining in the third instar larva and observed a similar gradient of GFP signals towards the posterior end of the ventral nerve cord (VNC). We have included this data in the revised manuscript (new Figure 5—figure supplement 1).

(5) The authors state that either 2nd or 3rd instar larvae were used for the optogenetic experiments. This may induce unnecessary variation in their assay and should be avoided. As natural variance exists in larvae regarding forward stride duration, the comparison of "on" state forward stride duration between control and experimental genotype is potentially not the best measurement of effect size. What is the difference between OFF and ON stage within the control and experimental genotype? In both cases stride duration decreases but there may not be a significant difference between the delta of the two genotypes. Thus, the observed effect may in part be due to "slower" animals in the control pool. The authors should discuss this more carefully.

We thank the reviewer for bringing up this critical issue. Indeed, the stride durations of larvae between the control and DFz2 knock-down are slightly different in the OFF condition, although this is not statistically significant. In addition, the effect size of Wave activation on mean stride duration is -0.14 (s) in control while -0.21 (s) in DFz2 knock-down, which we interpret as DFz2 knock-down resulting in stronger fastforward locomotion upon Wave activation. We have incorporated this note in the corresponding figure legends (new Figure 6; formerly Figure 5).

(6) While the study clearly provides convincing evidence for their model, the authors should tune down their conclusions in the discussion a little bit and highlight that parts of their discussion are speculative.

We have revised the discussion as suggested.

**Reviewer #2 (Recommendations For The Authors):**
Albeit the optogenetic behavioral experiments strongly support that the altered axonal projection affect normal locomotion, simultaneous labeling of Wave neurons in DFz2 KD animals with presynaptic markers would strengthen the conclusion of ectopic connection of the extended axon with other circuits.

Please see our response to your public review.

Figure 1 K+L, Figure 2H, I, Figure 3 F+G: many of the individual data points are not visible in the Whisker plot- changing their color would be useful to visualize them better.

We have changed the outline width of the box plots to make the individual data points visible.

Figure 1-Supplement 2: In addition to the comments in the public review- (a) the asterisk font size changes in the different panels, e.g. it is much smaller in G', (b) font size in some graphs/legends should be increased - in particular in E the hyphenated letters in the genotypes are so small rendering them almost illegible.

We have unified the font size to make them readable in the figure. We thank the reviewer for the suggestions.